# Memristive tonotopic mapping with volatile resistive switching memory devices

Alessandro Milozzi [1], Saverio Ricci [1] & Daniele Ielmini [1] ✉

To reach the energy efficiency and the computing capability of biological neural networks, novel hardware systems and paradigms are required where the information needs to be processed in both spatial and temporal domains. Resistive switching memory (RRAM) devices appear as key enablers for the implementation of large-scale neuromorphic computing systems with high energy efficiency and extended scalability. Demonstrating a full set of spatiotemporal primitives with RRAM-based circuits remains an open challenge. By taking inspiration from the neurobiological processes in the human auditory systems, we develop neuromorphic circuits for memristive tonotopic mapping via volatile RRAM devices. Based on a generalized stochastic device-level approach, we demonstrate the main features of signal processing of cochlea, namely logarithmic integration and tonotopic mapping of signals. We also show that our tonotopic classification is suitable for speech recognition. These results support memristive devices for physical processing of temporal signals, thus paving the way for energy efficient, high density neuromorphic systems.

Perception of information from the surrounding environment is a crucial task for animals to detect external stimuli and react to them. Light, sound, gravity, touch, and chemicals are converted into encoded spiking signals by dedicated apparatus and then interpreted by the brain[1]. Because of its high-energy efficiency and intrinsic error tolerance, the human brain provides inspiring novel paradigms to achieve better computational performance[2–6]. In this framework, the auditory system has gained strong attention due to its remarkable features: for instance, sound can reach our ears from all possible directions in space and can be perceived anytime when we are awake as well as we are sleeping. Moreover, sound processing is not performed by the spatial arrangement of sensory-afferent neurons such as in retinotopic or somatosensory maps, rather it is internally processed by the auditory system thanks to an internal representation of physical features[1,7,8]. As a result, the auditory system does not rely on the spatial position of the source such as in vision, where different light rays are focused on different sensors in the retina. Instead, the sound is processed via mechanical vibrations that are purely temporal signals. The frequency of natural sounds perceived by mammals usually spans from tens of Hz to tens of kHz, covering about three orders of magnitude. To classify these signals, there is a need for a spatial representation of this broad range of temporal features. The cochlea solves this task by realizing a tonotopic map of the incoming signals i.e., a mapping of different frequency components along logarithmically-spaced positions of the cochlear channel[1,9,10]. Emulating this kind of spatiotemporal signal processing through simple and scalable hardware remains an open challenge for neuromorphic computing[11].

Resistive switching memory (RRAM) devices have attracted strong interest for their ability to implement artificial neurons and synapses in high-density, energy-efficient artificial neural networks[12–17]. However, the main computing approach adopted in state-of-the-art systems relies on the spatial arrangement of neural elements, i.e., spatial coding. In these systems, the capability to capture the temporal component is introduced through complex auxiliary CMOS circuitry and sophisticated temporal encoding of the programming pulses, thus losing advantages in terms of area occupation, energy efficiency, and biological plausibility. This is because RRAM devices are used as static first-order memristors that are unable to directly cope with

[1]Dipartimento di Elettronica, Informazione e Bioingegneria, Politecnico di Milano and IU.NET, Piazza Leonardo da Vinci 32, 20133 Milano, Italy.
✉e-mail: daniele.ielmini@polimi.it

spatiotemporal signals, just playing the role of static memory for mapping weights of neural networks[18]. To address this limitation and enable device-level computation over time and frequency, it becomes imperative to explore innovative materials and methodologies within the increasing set of memristive devices. Such advancements hold substantial promise for enhancing spatiotemporal pattern recognition by exploiting the intrinsic dynamics of the device to capture the crucial temporal component that is otherwise missing. Moreover, temporal features generally cover broad scales, while actual demonstrations of neurons and synapses integration mostly operate on a limited linear scale due to physical limitations in the mechanisms of conductance change[19–21]. This is in contrast with the brain being capable of perception and classification of sound over a broad frequency range and in presence of noisy signals[22]. By leveraging the dynamic, stochastic response of volatile memristors, we demonstrate a device-level spatial mapping of temporal spike signals on a logarithmic scale, where, similar to biological systems, the device volatility contributes to the system ability to relax to a resting state, being spontaneously ready for a new computation. These characteristics serve as the fundamental ground for replicating the intricate audio processing functions executed by the human brain.

## Results

### Stochastic switching of volatile RRAM devices

To enable time and rate computation, we adopted volatile RRAM with one-transistor/one resistor (1T1R) structure. Figure 1a shows a schematic illustration of the device, where the select transistor allows to control the maximum current flow (see Supplementary Note 2 for MOSFET characteristics). The device relies on a switching layer made of hafnium oxide ($HfO_x$) interposed between two metallic electrodes. Figure 1b shows the device structure, including a silver (Ag) active top electrode (TE) and a bottom electrode (BE) made of carbon (C). The RRAM device is initially in a high resistive state (HRS) due to the low electrical conductivity of the $HfO_x$ layer. The application of a relatively large positive voltage between TE and BE results in the formation of a conductive filament (CF) made of migrated Ag atoms thus resulting in a set transition to the low resistive state (LRS). When the voltage across the device is removed, the CF spontaneously dissolves after a suitable retention time, bringing the device back to the HRS[23–25]. Thanks to the spontaneous dissolution of the CF, differently from non-volatile RRAM devices, our device does not need a reset phase thus it can operate

with unipolar voltages. Figure 1c shows the measured quasi-static current-voltage (I-V) curve of the RRAM devices, indicating the presence of a characteristic threshold voltage $V_{set}$ to initiate Ag migration and to build the CF and a hold voltage $V_{hold}$ to start the dissolution. The value of $V_{set}$ might stochastically change from cycle to cycle due to the continuous rearrangement of materials structure at the interfaces and in the switching layer[26]. The cycle-to-cycle $V_{set}$ variation generally obeys a normal distribution (see Supplementary Notes 3–4 where device-to-device variability is also reported). The $V_{set}$ distribution describes the probability for set transition by applying a pulse with a specific voltage amplitude and duration. High values of voltage amplitude compared to the median value of the distribution of threshold voltage result in a high probability of switching while low values of voltage amplitude result in a low probability of switching.

To characterize the switching probability of our RRAM devices in pulsed regime, we applied a train of voltage spikes reported in Fig. 2a with fixed pulse-width $T_{pulse} = 2.5\,\mu s$ and a total duration of $T_{window} = 25\,ms$. Figure 2b shows the response current and the evolution of the device conductance $G$. After an initial phase where the current is zero, corresponding to the device being in the OFF state, the device switches to the ON state after a number of spikes equal to $N_{set} = 38$ which is marked by the onset of a current response and a transition to $G = 8\,\mu S$. In this experiment, the waiting time between each spike was lower than the retention time, thus ensuring that the device remains in the ON state (see Supplementary Note 5). Figure 2c shows the current response for various spike voltage amplitudes, indicating that $N_{set}$ (hence the switching time $t_{ON}$) decreases for increasing spike amplitude (details about waveforms are reported in Supplementary Note 6). Figure 2d shows the average $t_{ON}$ as a function of voltage amplitude and spike frequency with fixed $T_{pulse} = 2.5\,\mu s$, indicating that $t_{ON}$ decreases at increasing voltage amplitudes and frequency. We can define the switching probability due to the application of a train of spikes as:

$$P_{switch} = \frac{N_{trains|ON}}{N_{trains}} \tag{1}$$

where, $N_{trains|ON}$ represents the count of applied trains that cause the device to switch to the ON state and $N_{trains}$ is the total number of applied trains ($N_{trains|ON} \leq N_{trains}$). It is important to notice that this

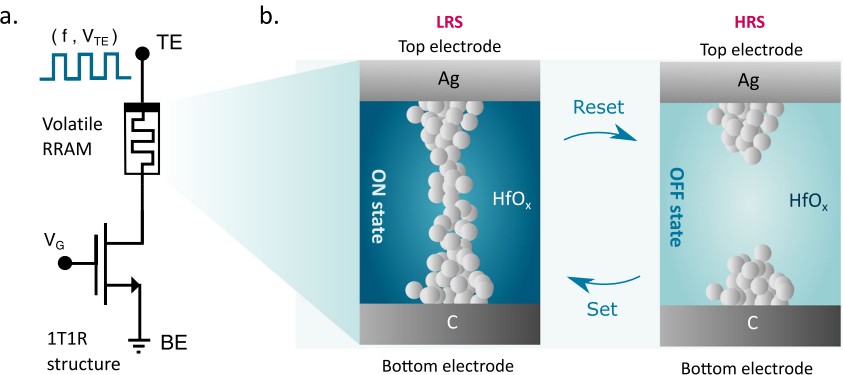

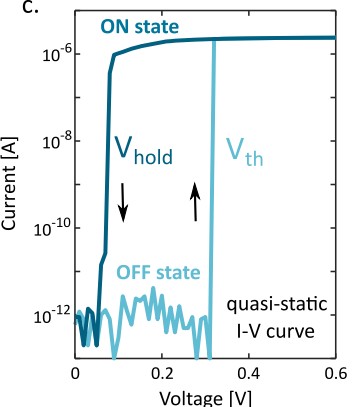

**Fig. 1 | Ag-based resistive random access memory. a** 1T1R structure used in the experimental characterization: the programming pulses are applied to the top electrode (TE) while the bottom electrode (BE) is grounded and used to read the current. The transistor is placed in series to obtain better control of the maximum flowing current. (see Supplementary Note 1 for the experimental setup) **b** Schematic description of the arrangement of the silver atoms inside hafnium oxide: in the low resistive state (LRS) or ON state, silver atoms build a bridge between electrodes resulting in a high value for conductance. In the high resistive state (HRS) or OFF state, there is no conductive path between the top electrode and bottom electrode resulting in a low conductance value. **c** Quasi-static I-V curve of the device: when the applied voltage is above the threshold voltage, the device switches on and moves to LRS. When the applied voltage is lower than hold voltage, RRAM cell relaxes to HRS.

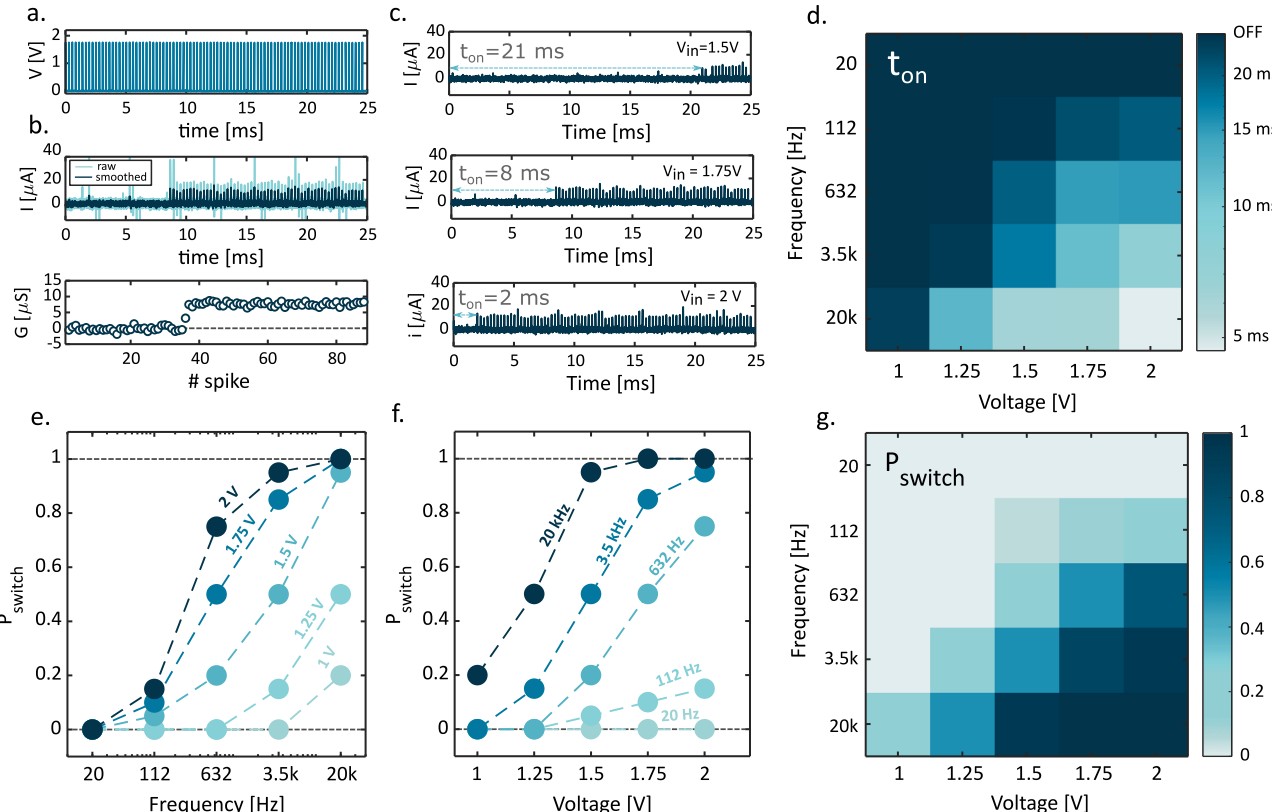

**Fig. 2 | Switching time and switching probability for different operative conditions. a** Voltage pulse train for programming: a train of equally spaced voltage pulses of duration $t_{pulse} = 2.5\,\mu s$ and a fixed time window duration of $T_{window} = 25\,ms$ is applied to the device. **b** Current response to voltage pulse train: initially, the device is in the OFF state and the current is zero. After some spike, the RRAM switches on, and the current increases. Respective conductance values associated with spike number show a step. **c** Time to switch on: the time needed for the device to turn on is depending on the voltage amplitude of the train. Increasing the voltage, the time decreases. **d** Time to switch on heat map under different operative conditions: Changing the frequency of the spikes and the voltage amplitude of the train, the time to switch on changes. In particular, the minimum of time is reached for the maximum tested voltage and frequency while the device remains in the OFF state for minimum tested voltage and frequency. **e** Switching probability versus frequency: Increasing the frequency of the train of pulses, the probability of switching increases due to a larger number of spikes applied to the device i.e., larger number of trial to switch it on. **f** Switching probability versus voltage amplitude: increasing the voltage amplitude of the train of pulses, also the switching probability increases. **g** Switching probability heat map for all the combinations of voltage and frequency summarizing the experimental characterization.

definition of switching probability does not refer to the single pulse, but rather to a train of pulses with amplitude $V$, frequency $f$, and duration $T_{window}$, representing a more comprehensive and generalized framework for addressing switching probability. Figure 2e shows that $P_{switch}$ increases by increasing the spiking frequency, while Fig. 2f shows that $P_{switch}$ increases by increasing the applied voltage. Figure 2g summarizes the dependence of $P_{switch}$ on $f$ and $V$, respectively. Note that the frequencies on the $x$-axis of Fig. 2e and on the $y$-axis of Fig. 2g are logarithmically spaced, spanning 3 orders of magnitude, in analogy with the pitch tonotopic classification in the human cochlea.

## RRAM circuit for frequency sensing

Based on the spiking frequency properties of the device, Fig. 3a shows the RRAM circuit to provide the tonotopic sensing of the auditory signal frequency. In this circuit, RRAM devices have separate TE and a common BE to collect the summation current from all devices based on Kirchhoff's law. The gate voltage is common for all devices, thus ensuring that the current is approximately the same for each device in the ON state due to transistor channel saturation. The spike trains applied to different TEs have the same frequency while the voltage amplitude $V_{TE}$ decreases from one TE to the next one, e.g., TE voltages $V_1 = 2\,V$, $V_2 = 1.5\,V$, and $V_3 = 1\,V$ are applied to the three RRAM devices in Fig. 3a. Based on Fig. 2, the application of a signal at relatively low frequency causes the switching of only a small fraction of devices in the

range of high $V_{TE}$, while an input signal with high-frequency causes the switching of a large fraction of devices, including those biased at relatively low voltages.

Figure 3b shows a trace example of the measured response current for the circuit of Fig. 3a, where spiking trains were applied with increasing frequencies from 20 Hz to 20 kHz. Based on the maximum measured current, it is possible to infer the number of devices in the ON state thanks to the compliance current $I_c$ of the select transistor. Results in Fig. 3b indicate that in the reported experiment, for $f = 20\,Hz$, none of the devices can switch within the experimental time window of 25 ms. As we increase the frequency of spikes, the number of devices in the ON state increases, reaching the maximum of 3 devices in the ON state for $f = 20\,kHz$. In this case, it is possible to identify three distinct steps in the current trace, each corresponding to the switching on of a device. Figure 3c shows the experimental histograms of the number of devices switching to the ON state for a specific spiking frequency: Being normalized histograms, it is possible to interpret the y-axis as the probability $P_{N,ON}$ of observing a specific number of ON-state devices (shown on the x-axis) for a particular frequency of input train. Figure 3d shows the average number of devices in the ON state as a function of the input train frequency. Note that the number of devices in the ON state increases linearly with the logarithm of the input frequency. Such a logarithmic dependence of the frequency sensitivity is the key point for processing audio signals in the auditory system from the environment[1]. In the cochlea, in fact,

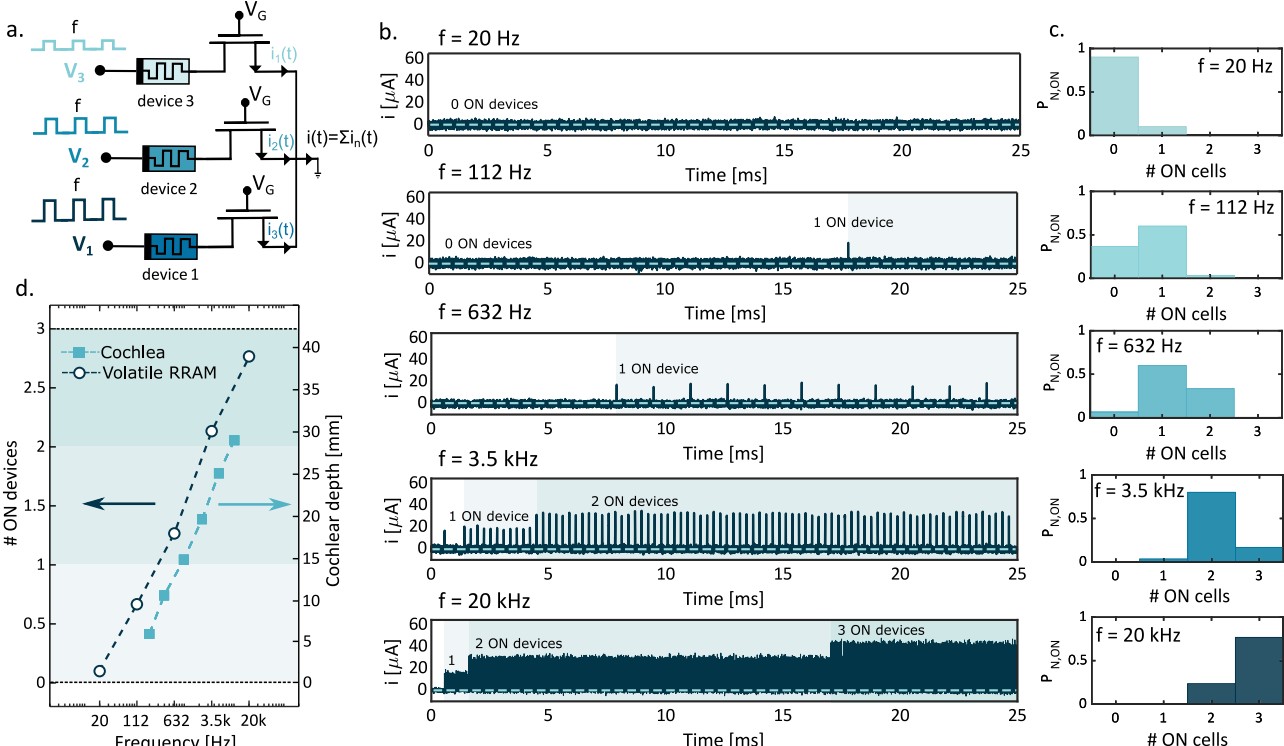

**Fig. 3 | Experimental demonstration of RRAM circuit for frequency sensing.**
**a** Schematic of the arrangement of parallel cells: each device has its own TE while the BE is in common, where the sum of the currents from devices is collected. The gate voltage is the same for all the MOSFET and it is equal to $V_G = 1\,V$. The spike trains applied to the top electrodes have the same frequency while the voltage amplitude is scaled for each device from higher to lower. **b** Examples of filtered temporal current traces of i(t) collected from BE in the experiment: when f = 20 Hz, none of the devices switch on. Increasing the frequency, also the number of ON devices increases. It is possible to see that for f = 20 kHz the devices switch on at different times, resulting in 3 steps in the current traces. (see Supplementary Note 9 for raw current traces.) **c** Normalized histograms of the number of ON devices at

different frequencies: Increasing the frequency, the peak of the histogram moves to the right due to an increase in the number of ON cells. **d** Linear mapping of log-spaced frequencies in our systems and in the cochlea after Zwislocki[28]: Applying trains of different frequencies to the system of parallel devices, the number of ON cells is proportional to the logarithm of the frequency thanks to the logarithmic dependence of switching probability (dark blue curve). The same behavior of mapping the audio frequency range in a linear space has been observed in the cochlea (light blue curve). Here, logarithmically spaced frequencies from 20 Hz to 20 kHz are mapped into different linearly spaced locations from the apex of the cochlea.

different frequencies from 20 Hz to 20 kHz are mapped into linearly spaced distances from the apex of the cochlea as reported in Fig. 3d[27,28]. This highlights the similarity of the RRAM-based frequency sensing circuit of Fig. 3a to the biological cochlea.

These experimental results were described by a probabilistic model reported in Supplementary Note 10 and validated in Supplementary Note 11. The model enables the simulation of large-scale networks and is utilized in the following sections to simulate the logarithmic integration and tonotopic mapping in bioinspired neuromorphic systems where the variability of device-to-device switching probability is included (see Supplementary Note 8). Nevertheless, the model applies to any RRAM device by adjusting the model parameters, e.g., the threshold voltages and their distributions, thus providing a general simulation tool for stochastic computing with resistive switching devices.

**Cochlea-inspired tonotopic sensing of audio frequency**
Figure 4a schematically illustrates the human auditory system where the acoustic wave reaches the tympanic membrane and the cochlea. Along the cochlear channel, different frequencies of the acoustic wave are detected at different positions by the hair cells, which are specialized biological strain detectors[29]. The stimulation of hair cells causes the mechanical opening of ion channels, thus enabling the flow of a small ionic current converting mechanical stimulation into an electrical signal[30,31], which eventually propagates to the brain through auditory nerves. High frequencies (up to 20 kHz) are

detected in the initial part of the cochlea, while low frequencies (around 20 Hz) are detected in the deepest region of the cochlea i.e., the center of the spiral. The intermediate values are logarithmically spaced through the length of the cochlear channel[28,32]. The cochlea thus allows for a spatiotemporal processing capable of mapping temporal signals in different spatial coordinates of the cochlear channel. Such a tonotopic map of frequencies was experimentally demonstrated by von Békésy, worth the Nobel prize for medicine in 1961[33]. Figure 4b shows the calculation results of a model derived after Zwislocki[28]: the spectral amplitude response shows a peak for a particular frequency, thus enabling frequency detection on a logarithmic scale.

Emulating the processing of the auditory system, Fig. 4c shows the schematic of a RRAM circuit that enables tonotopic mapping of different frequencies which we refer to as memristive tonotopic map (MTM), where we completed a system of parallel volatile RRAMs with a XOR gate comparing the output voltages in each pair of RRAM devices. As in the previous experiment, the trains applied to different top electrodes have a common frequency, corresponding to the input signal frequency, while the applied voltage $V_{TE}$ decreases from the highest $V_1$ to the lowest $V_N$. As a result, device $i+1$ is sensitive to higher frequencies compared to the device i, with index $i = 1, 2, ..., N$. This property becomes evident when examining the calculated switching probability $P_{switch}$ shown in Fig. 4c: we show this behavior in six cells with increasing $V_{TE}$ values, derived from our probabilistic model, which was calibrated using experimental

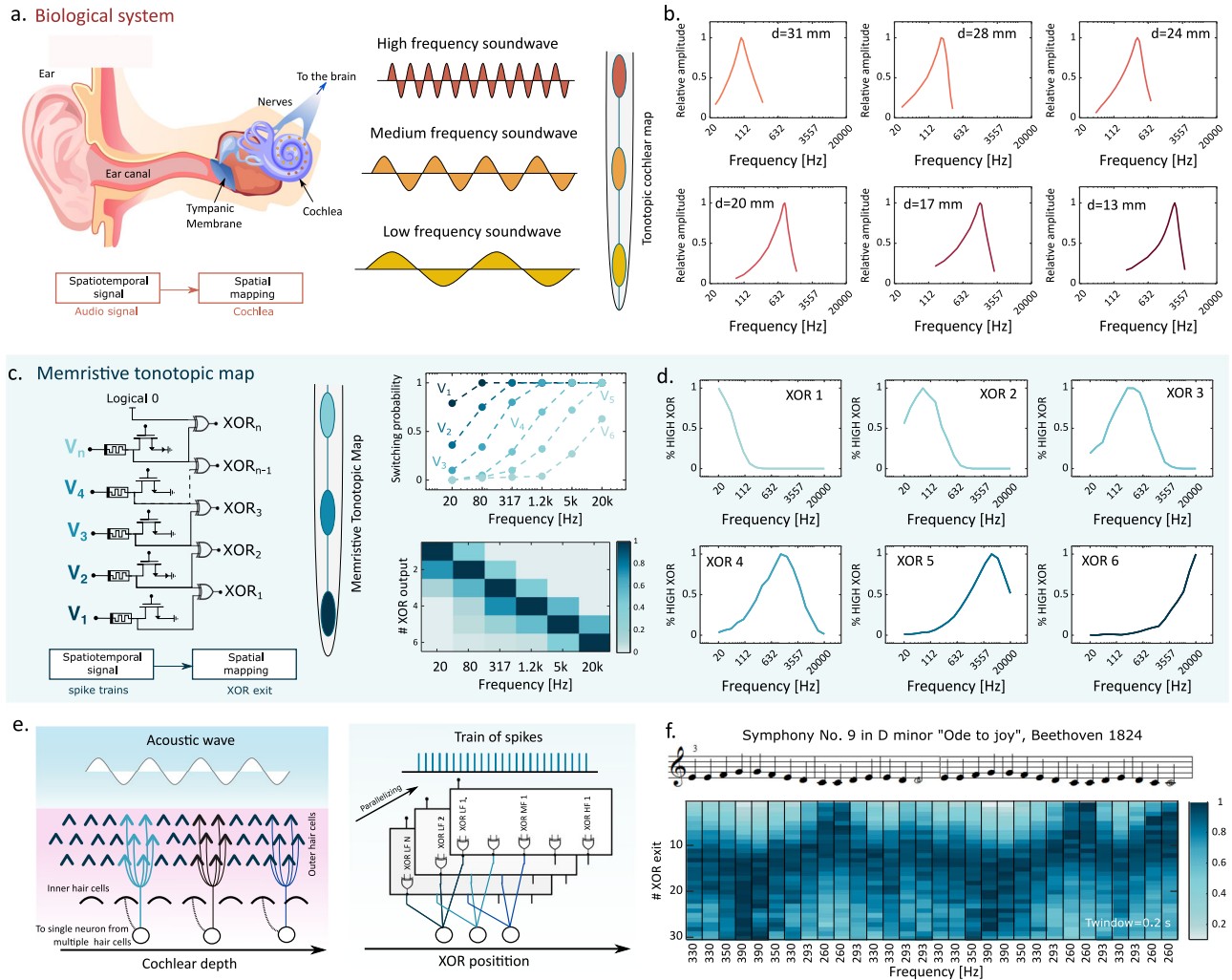

**Fig. 4 | Tonotopic mapping of spatiotemporal signals to emulate cochlea processing. a** Schematic representation of the human ear: the acoustic wave entering the ear canal reaches the tympanic membrane. Here, mechanical oscillations in the air are propagated to the lymph of the cochlear channel. Different oscillation frequencies are detected by hair cells in different locations of the cochlea. Red dots are regions in which higher frequencies are detected while going towards yellow dots, lower frequencies are distinguished in the inner part of the cochlea. **b** Relative response of hair cells for different distances from the cochlea apex, model from[28]: each region shows a peak for a particular frequency, enabling frequency detection. **c** Memristive tonotopic map (MTM): each cell is tuned with a different spike train voltage amplitude from the maximum $V_1$ to minimum $V_n$. The P(f) plot shows for $n = 6$ how this voltage arrangement results in different probabilities to switch on the device. Low-index devices (i = 1) are low-frequency sensitives while high-index devices are high-frequency sensitives (i = 6). XOR logic

enables the detection of the location of transition between ON and OFF devices. #XOR output versus frequency plot shows that for each frequency, the maximum response is related to a specific XOR i.e., specific spatial position. **d** Normalized response for different XOR locations: similarly to the cochlea, each XOR shows a peak in the response to a specific frequency. Moving to a higher index XOR, also the peak of the frequency moves to higher values. **e** The analogy between hair cells arrangement and stochasticity mitigation technique in MTM: in the cochlea, many hair cells are present and there is redundancy to mitigate damaged cells and stochasticity. Following the same approach, we put in parallel N memristive tonotopic maps fed by the same train to obtain an average response. **f** Response of the system to "Symphony No. 9 in D minor": with $n = 30$ cells for $N = 500$ parallel MTM, we can reach higher frequency resolution and we can see how the position of the most active XOR changes following the music sheet also for a moderate span of frequencies.

characterization data (see Supplementary Note 11). Specifically, as $V_{TE}$ increases, the probability of switching to a lower frequency also increases. The role of the XOR gate is to identify the boundary between the last ON device and the first OFF device (see Supplementary Note 12). Figure 4c also shows a matrix plot of the simulated average XOR output as a function of frequency, indicating that the response of each XOR gate is maximum for a specific frequency. This behavior is further highlighted in Fig. 4d, showing the normalized simulated response for each XOR output as function of train frequency. The first XOR gate has peak activity around $f = 20$ Hz, while XOR gates at higher orders respond at higher frequencies. The maximum frequency of the audio range ($f = 20$ kHz) is detected in the last XOR.

This approach can be further optimized by emulating the hair-cell redundancy in the biological system: as depicted in Fig. 4e (left), hair cells in the cochlea are on average 15.000, working in different locations but also in parallel at the same location thus introducing redundancy to mitigate the effects of stochasticity and/or malfunctioning hair cells. The same approach can be emulated by introducing a higher number of parallel RRAM devices in the tonotopic circuit to provide a better averaging as reported schematically in Fig. 4e (right). To support this latter approach, we simulated a larger network with $n = 30$ cells in the circuit of Fig. 4c and 500 parallel RRAM devices for each frequency. As an audio sample, we choose the "Finale" of Symphony n°9 in D minor by Beethoven due to its reduced tones (frequencies) spanning from C note (260 Hz) to G note (390 Hz).

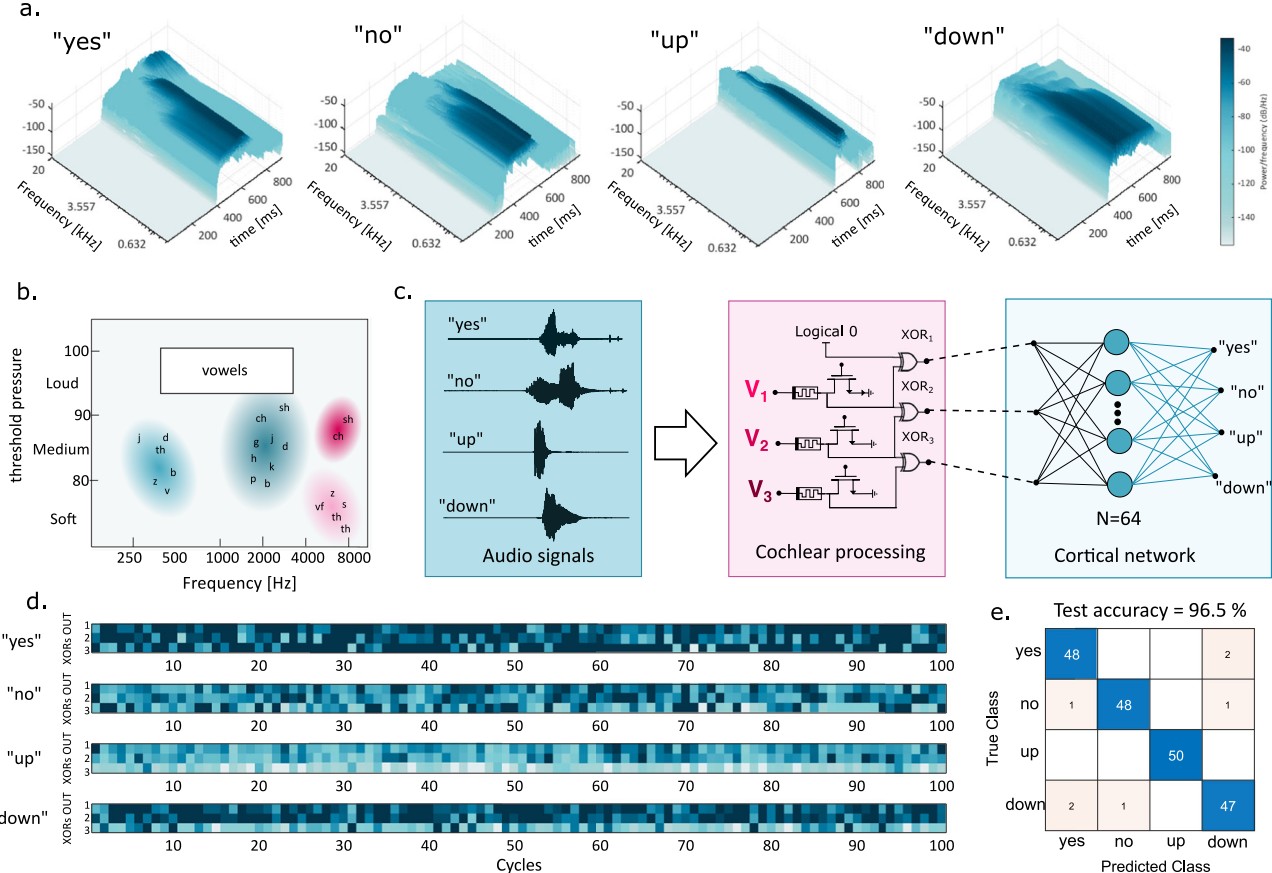

**Fig. 5 | Speech recognition from the tonotopic map. a** Spectrograms for different spoken words: each combination of phonemes results in different frequency compositions and different amplitudes of power spectrum. **b** Schematic map of different phonemes as a function of their principal frequency component and sensation level for hearing (adapted from ref. 36). **c** Artificial auditory system: audio signals of different words are processed with simple analog-to-spike conversion.

Trains of spikes reach the cochlear processing layer made with a MTM and its output is fed to an FF-neural network for classification. **d** Outputs of XORs for different spoken words for 100 different cycles. For each cycle, the selected word is randomly picked from the set of available audio samples. **e** Confusion matrix for the classification of spoken words.

Figure 4f shows the simulated response of the circuit, clearly indicating the change of the active XOR following the behavior of the music sheet.

## Interpretability of tonotopic map: speech recognition

In the biological auditory system, the electrical signals generated by the cochlea in response to the incoming audio signal need to be interpreted by the auditory cortex[34]. Here, a classification process takes place to recognize the sound, discriminating the tweet of a bird from the sound of the flowing water of a river. The human brain, moreover, shows the capability of speech recognition i.e., the capability of attributing meaning to a particular combination of phonemes[35]. To demonstrate such capability in our memristive auditory system, we simulated the task of speech recognition implemented with our MTM. We selected a set of 4 words that carry logical information ("yes", "no") or spatial information ("up", "down") spoken by a single person and repeated 20 times. Figure 5a shows the spectrograms of audio traces for these four words: one can notice different time durations, e.g., "up" is shorter than "down", and different frequency spectra, e.g., the high frequencies in "yes" due to the letter "s". Also, note that phonemes differ not only in terms of frequencies but also regarding the pressure that they exert on hair cells. Figure 5b adapted from ref. 36, shows a map of phonemes that are located depending on their frequency and threshold pressure, highlighting the features used by biological systems for classification. Vowels span a broad spectrum of frequencies and are linked to a

higher threshold pressure, such as the higher amplitude of the sound "ye" in "yes" and "o" in "no". The system for speech recognition is shown in Fig. 5c, with the MTM for $n = 3$. We kept this value as low as possible to showcase the capabilities of our system with a minimal number of elements, thus gaining in power consumption and occupied area (see Supplementary Note 13 for results for different n values and different numbers of cortical network neurons). Spike trains are applied to each TE based on the raw normalized audio trace with the analog-to-spike (A2S) conversion reported in Supplementary Note 14. The conversion operates with different thresholds for each channel to capture different pressure levels i.e., amplitude of the signal, thus providing an additional feature for the recognition task. The three outputs of the XOR gates are submitted to a classical feedforward neural network that performs the same role as the auditory cortex of the biological brain. We selected $N = 20$ for the MTM parallelization to obtain an average response to the applied trains. Figure 5d shows the output values of MTM for each XOR for 100 trials for each word. At every cycle, the audio sample of the selected word is randomly chosen from the set of available samples. It is possible to recognize the different patterns, where, e.g., "yes" corresponds to a higher average activity while "up" corresponds to a lower average activity. This behavior can be attributed to the spectrograms of Fig. 5a, where "up" displays a shorter duration and medium-frequency composition whereas "yes" displays a longer duration and high-frequency components. We used the first 50 examples of MTM output traces to train the neural network and

the last 50 examples for inference. Figure 5e shows the confusion matrix for the inference, reaching an accuracy of 96.5%.

## Discussion

By exploring the building blocks of sensory biological systems, it is possible to identify the features that are responsible for the exceptional computation capabilities of the brain. Implementing these paradigms in hardware, however, requires new technologies and methodologies. Through CMOS technology system-level approach, simulation of the functional behavior of biological mechanisms can be achieved, however at the cost of a large area occupation, a complex design, and a high power consumption (an indicative comparison is provided in Supplementary Note 15). Furthermore, these systems pose challenges in terms of interpretability, as the biological mapping and meaning are often lost[37]. This difficulty makes it challenging to implement powerful computational neuroscience models in hardware, thus hindering the potential to attain the efficiency and capabilities of the biological brain[38].

Memristive devices mitigate these issues thanks to their high scalability and low power, as well as the capability to directly compute within the memory for reduced latency and energy consumption. Memristive devices display 2 terminal devices and tunable conductance, thus providing a realistic hardware description of synaptic plasticity in the brain, paving the way for the emulation of biological neural elements. Major research efforts have gone in this direction, where the memristive element is used as a static first-order memory element to store the synaptic weight of an artificial neural network[39]. In this framework, memristors can act as accelerators of hardware neural networks within the context of in-memory computing[40,41]. Other research directions aim at the exploration of biologically plausible paradigms using memristors as synaptic dynamic elements. The main features of biological mechanisms such as spike-timing dependent plasticity (STDP) and Hebbian learning have been demonstrated through these approaches[42-44]. However, the full potential of memristor devices can be exploited by moving the computation to the device level, thus minimizing the need for external circuitry[45], building through a bottom-up approach, explainable and biologically plausible systems.

In our work, the computation in memristors relies on the probability of switching i.e., the probability of transmission of the information in a volatile memory, dealing with bursts of spikes rather than individual spikes. Biological systems, in fact, operate through an ensemble of probabilistic elements that together perform computation in a stochastic way[46]. Moreover, biological networks make unreliable elements sufficiently reliable by working with spike bursts as the units of neural information[47]. Thanks to this approach, we have shown that it is possible to perform a stochastic integration of the number of spikes, where the integrable range can be controlled by the voltage amplitude of spikes without the need for large capacitances. This result enables a new approach for spiking networks, capable of enlarging the space of computation to a logarithmic range of times and/or frequencies. Moreover, as the brain does not rely on a single synapse, we are not relying on a single device to properly mitigate the stochastic variation, using multiple parallel devices. Our methodology, which can map 3 orders of magnitudes of temporal features into a linear space of voltage, has a general validity beyond audio signals, with applications ranging from tactile to visual sensors[48,49]. Also, there is a large margin for improving the system by increasing the number of devices in the single MTM and increasing the number of parallel MTMs to perform a more challenging task, such as recognizing more phonemes and even complete words. However, large-scale simulations go beyond the scope of this work, which is mainly focused on the proof-of-concept demonstration of the tonotopic classification of audio signals by memristive networks. Nevertheless, our devices promise good energy scaling even in large systems: when the device is in the

OFF state, the resistance value is in the range of tens of TΩ, and no relevant current is flowing in the device. Figure 1c shows that the OFF-current is below the resolution of the instrument in the order of picoampere (pA). Referring to the same figure, it is possible to see that when the device is in the ON state the current $I_C$ flowing in the 1T1R series is limited by the transistor. In this work, the value of $I_C$ has been chosen high enough to be properly readable through a low-impedance channel of the oscilloscope (current resolution in the order of $1\,\mu A$). This value also provides a fast estimation of the energy consumption of a single spike through the ON device, given by:

$$E = V_{TE}I_C t_{pulse} = 1V \times 16\,\mu A \times 2.5\,\mu s = 40\,pJ \qquad (2)$$

However, note that the device can be switched on with much lower currents (see Supplementary Note 16) down to $I_C = 10\,nA$, thus providing excellent scalability of energy consumption. The ultra-low energy operation also allows for efficient parallelization, accommodating high values of n and N parameters for MTM. Additionally, the presented system achieves logarithmic integration through a probability mechanism, thus eliminating the need to supply energy for charging a capacitance with each spike or for integration operations. Thus, the device reduces the number of spikes, being effective in consumption just after its switching i.e., when conductance is in the LRS.

The device-level approach also exploits the volatility of the memristive devices, which allows the spontaneous relaxation to a ground energy state of the system where all devices are OFF. As a result, after sufficient time dictated by the RRAM retention, the system becomes ready again for a new computation without any need for repeated initialization, thus leading to a substantial reduction of energy consumption and system complexity. This property is also fundamental to emulate the asynchronous computation of the biological nervous system since the arrival of the signal serves as the triggering input for computation while the absence of the signal lets the system return to the ground state by spontaneous relaxation[1,50]. Furthermore, since the reset phase is not required for our RRAM, the device can be operated with unipolar voltages, thus providing a significant advantage in the complexity of programming circuits and area scaling of the selector device. Although the area scaling of the 1T1R structure is not the focus of this study, it is useful to highlight certain features of the presented RRAM device that may offer insight for future developments. Specifically, our device exhibits the capability to reduce the crucial parameters of programming voltage and programming current in scaling 1T1R structures, in compliance with the requirements for the integration of RRAM in the most recent technological nodes[51]. Additionally, it is important to emphasize that, due to its volatility, the device does not require bipolar voltages for resetting since it can operate in a unipolar manner. This characteristic could also potentially pave the way for a transition to a bipolar junction transistor (BJT) selector, similar to the reported phase-change memory (PCM) technology with related advantages[52].

In summary, this work presents the neuromorphic circuits for spatio-temporal signal processing with volatile RRAM relying on a device-level approach. We demonstrate the implementation of the main neuromorphic primitives for audio signal processing in the cochlea, namely logarithmic integration and tonotopic mapping of temporal information. Our tonotopic transformation is suitable for speech recognition mimicking the biological counterpart, preserving biological plausibility and explainability. These results have a general validity beyond the audio signal processing thus supporting memristive device for hardware processing of temporal signals with logarithmically spaced features, enlarging the set of available neuromorphic primitives necessary to reach the energy efficiency, error tolerance, and high integration density promised by neuromorphic computing paradigm based on memristive devices.

## Methods

### Devices fabrication

The volatile resistive devices presented in this work are co-integrated on with Si-based transistors fabricated with standard CMOS technology. The bottom electrode is a graphitic carbon pillar with a diameter of 70 nm connected to the transistor drain. The 5 nm of hafnium oxide ($HfO_x$) switching layer and the 100 nm silver (Ag) top electrode are sequentially deposited by e-beam evaporation without breaking the vacuum, within a monitored pressure of $3 \times 10^{-6}$ mbar to carefully tune the $HfO_x$ stoichiometry and the $Ag/HfO_x$ interface quality.

### Devices characterization

All electrical characterizations and experiments are carried out in probe station by using rhodium-plated tungsten needles for contacting. Semiconductor parameter analyzer Agilent HP4156C is used for the quasi-static characterization of the devices. Dynamic properties, as well as the experiments, are studied using an AimTTi TGA12104 Arbitrary Waveform Generator and a Tektronik MSO58 Oscilloscope for the acquisition.

### Switching probability measurement protocol

In the presented measurement, the train amplitude and frequency were set randomly in each cycle to avoid correlation effects. We tested every possible combination of selected voltage and frequency for the desired number of cycles. Supplementary Note 7 schematically reports our measurement protocol.

### Simulations

All the numerical simulations concerning the switching probability and MTM for cochlear-sensing and speech recognition are carried out on MATLAB R2022b with our developed models. Neural network training and inference for interpretability of MTM results are performed with MATLAB Statistical and Machine learning toolbox.

### Analog-to-Spike conversion

For our study of speech recognition, the analog audio signals are processed with an analog-to-spike conversion algorithm. The information about the amplitude of audio signals is captured thanks to 3 different thresholds used to generate 3 different pulse waveforms that are supplied as input signal to the memristive tonotopic map circuit (see Supplementary Note 14 for the block diagram of analog-to-spike conversion and conversion traces examples).

## Data availability

All data that support the findings of this study are provided within the paper and its Supplementary Material. All additional information is available from the corresponding authors upon request.

## Code availability

The code used for simulations is available from the corresponding author upon request.

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

## Acknowledgements
This article has received funding from the European Union's Horizon 2020 research and innovation program (grant agreement No. 899559).

## Author contributions
A.M. and D.I. conceived the work direction and experiments. S.R. fabricated the volatile memristorsand set up the testing protocols. A.M. and S.R. performed the device characterization andexperiments. A.M. implemented stochastic model and system simulations. A.M., S.R. and D.I. wrotethe manuscript. D.I. supervised the project overall.

## Competing interests
The authors declare no competing interests.
