## [Peer Review File · Nature Communications]

Memristive tonotopic mapping with volatile resistive switching memory devicesREVIEWER COMMENTS

Reviewer #1 (Remarks to the Author):

General comment:

In this manuscript, the authors have developed neuromorphic circuits for memristive tonotopic mapping using volatile RRAM devices and have demonstrated the key aspects of cochlear signal processing. The paper is well-organized, and the main findings are both compelling and intriguing. I believe this work will have broad appeal and I am pleased to recommend the publication of this manuscript. However, I kindly request the authors to address the following points to further enhance the quality of the manuscript:

1. In the Introduction section, particularly in the second paragraph, the authors introduce both the advantages and development limitations of RRAM in bionic simulation. However, the relationship between the points presented and the narrative logic can be clearer. The authors state that "RRAM is capable of directly computing time and frequency," yet contemporary RRAM-based bionic demonstrations often require intricate peripheral circuits to achieve temporal encoding. The reasoning behind why current demonstrations operate in a linear scale is not well articulated. Additionally, the rationale for selecting volatile memristors to simulate the human cochlea could be more explicitly stated. More logical and unambiguous statements should be updated to improve readability.
2. Given the significance of the C2C (Cell-to-Cell) and D2D (Device-to-Device) properties of the basic volatile memristors in the subsequent RRAM circuit for frequency sensing, it is imperative to provide more comprehensive electrical characterization results. While the supporting information includes a presentation of only 20 cycles I-V sweep, a more thorough investigation, e.g., overall cycle stability and detailed pulsed test results, is recommended.
3. The individual transfer property of the adopted transistor should be provided.
4. Several abbreviations lack of specific explanation when first appear.
5. In Figure 2, the authors applied 1~2 V voltage pulses to switch the device ON. However, in Figure 1c, it is evident that the necessary threshold voltage (V_{th}) for switching is less than 0.4 V. This prompts the question of why a larger pulse voltage is required for switching, especially considering the relatively low V_{th} indicated in the previous figure. Furthermore, the switching probability under 1 V bias appears to be small, even when subjected to high-frequency pulse stimulation. Clarification regarding this apparent discrepancy is necessary.
6. The specific meaning of the y-axis in Figure 3c should be clarified to ensure a clear interpretation of the data. In addition, an explanation is required for the discrepancy between Figure 3b and Figure 3c, where none of the devices switch ON at a frequency of 20 Hz in Figure 3b, but a small probability of one device switching ON is observed in Figure 3c.
7. According to Kirchhoff's law, an increase in the number of ON-state devices typically results in a larger output current. However, a curious observation arises in Figure 3b, where the current levels for the cases of 1 device ON and 2 devices ON appear to be similar, except for the instance at $f = 20$ kHz. The authors should provide an in-depth analysis of the factors contributing to this peculiar behavior.
8. When considering the actual situation, I recommend using the term 'specific' rather than 'average' to describe the number of devices in the ON state as a function of the input train frequency in Figure 3d.

Reviewer #2 (Remarks to the Author):

The authors present innovative neuromorphic circuits for spatio-temporal signal processing

using silver-based volatile ReRAM. They successfully emulate the audio signal processing system in the human cochlea, achieving the tonotopic mapping of temporal information and classification of audio signals based on their frequencies. The work is commendably innovative and well-supported with data. However, it is regrettable that the speech recognition aspect relies on simulation rather than real memristor crossbars. Several concerns should be addressed before fully endorsing this work.

- 1, Clarification is needed regarding the definition of "Pswitch" in line 13 on page 4. The numerator in the equation appears to be $N_{\text{train}} - N_{\text{train}/\text{on}}$. This adjustment would render the observed trend of increasing Pswitch with higher applied voltage more reasonable.
- 2, Figure 2c should provide the pulse waveform to validate that the waiting time between each spike remains lower than the retention time.
3. It is advisable to include the variation of P switch in Figure 2 to facilitate comprehensive probabilistic studies. The impact of the variance observed in these measurements is noteworthy, and given its inherent presence, addressing this factor is essential.
4. In the experimental demonstration of the RRAM circuit for frequency sensing (Figure 3b), it's challenging to discern the number of devices switched on for temporal traces with frequencies exceeding 3.5kHz. A more discernible plotting method should be considered.
- 5, In the section of speech recognition (figure 5c), three XOR outputs may be not sufficient for extracting audio features, more than 5 outputs are recommended.
6. In the section of speech recognition (figure 5), only frequency signals are considered while the amplitude of audio signals is disregarded. the duration of distinct frequencies significantly influences the output of the XOR results. This temporal aspect constitutes another crucial factor that necessitates consideration.

Overall, the author's contribution is significant, though the concerns outlined above deserve attention to augment the credibility of their findings.

Reviewer #3 (Remarks to the Author):

The authors describe a RRAM-based circuit that could operate as an audio signal processing system. The RRAM devices used in this work are two terminal Ag/HfOx memristors. Probabilistic model of the RRAM is developed based on measurements of device switching probability versus pulse frequency or voltage amplitude. The manuscript discusses several advantages of this system, such as the unique capability in interpreting audio frequency signals and doing tonotopic mapping.

I believe the performance and capabilities of the presented system are impressive, even though I have some comments/questions that should be addressed.

1. Novelty at the system level is obvious, which is about the unique audio signal processing functionality. However, novelty at the device level seems not clear to me. First, the memristor structure and device mechanism in this work (Fig. 1) is nothing different compared to most other well-established memristors. Second, the switching probability of a memristive device versus pulsing frequency and voltage amplitude have both been studied in previous literature [such as Nanoscale 5, 5872–5878 (2013); Nat. Commun. 12, 5710, (2021)]. The

trend of switching probability provided in this work (Figs. 2-3) is thus well expected. The author should pay some attention in explaining about their novelty at the device level, otherwise the need of having Figs 1-3 for discussing what they found about the RRAM device is undermined.

2. The transistor in the 1T1R structure keeps a RRAM being in a volatile state instead of a non-volatile state, eg. the CF spontaneously dissolves after a suitable retention time. How important is the this “volatile” property in realizing the following audio signal processing applications?

3. Clearly the raw experimental data in the middle panels in Fig. 3b shows occasional spikes could be at high current level (~20-40 uA) due to the stochastic device behavior. If one high spike appeared, how could the authors determine if only one device is ON at this moment instead of interpreting that two devices are ON? It seems interpreting over single event is less reliable than interpreting the statistics over ensemble of multiple events. Will utilizing the stochastic property of RRAM in hardware require more energy due to the need to collect ensemble behaviors of a RRAM, as opposed to relying on a single event of a RRAM? The energy consumption could scale up quickly and become problematic if the number of RRAM devices increases.

4. Implied in the discussion is that the motivation for using memristive devices to replace CMOS technology system-level approach in mimicking biological functionalities are mainly due to advantages in power consumption, small area occupation, unique functionalities and so on.

I must ask where in this paper is evidence that this RRAM device does not suffer from the same challenges? In other words, the author should provide some direct comparison or benchmarking of power consumption and circuit footprint with CMOS technology and make some strong argument how RRAM in this work is advantageous over CMOS to better support their discussion.

5. Can author be more specific about what part is done by experiment and what part is done by simulation in all their application demonstrations (Figs. 4-5)? More details on how the measurements in the application demonstrations are performed should be provided.

Reviewer #1:

General comment:

In this manuscript, the authors have developed neuromorphic circuits for memristive tonotopic mapping using volatile RRAM devices and have demonstrated the key aspects of cochlear signal processing. The paper is well-organized, and the main findings are both compelling and intriguing. I believe this work will have broad appeal and I am pleased to recommend the publication of this manuscript. However, I kindly request the authors to address the following points to further enhance the quality of the manuscript:

We thank the Referee for the positive feedback on our work and very much appreciate the Reviewer's critical suggestions to improve the manuscript. Our responses to the comments are as follows.

1. In the Introduction section, particularly in the second paragraph, the authors introduce both the advantages and development limitations of RRAM in bionic simulation. However, the relationship between the points presented and the narrative logic can be clearer. The authors state that "RRAM is capable of directly computing time and frequency," yet contemporary RRAM-based bionic demonstrations often require intricate peripheral circuits to achieve temporal encoding. The reasoning behind why current demonstrations operate in a linear scale is not well articulated. Additionally, the rationale for selecting volatile memristors to simulate the human cochlea could be more explicitly stated. More logical and unambiguous statements should be updated to improve readability.

We thank the Referee for highlighting these points in the Introduction section. Following the Reviewer's suggestions, we have rewritten the second paragraph of the Introduction section to improve clarity and readability (pp. 2-3).

2. Given the significance of the C2C (Cell-to-Cell) and D2D (Device-to-Device) properties of the basic volatile memristors in the subsequent RRAM circuit for frequency sensing, it is imperative to provide more comprehensive electrical characterization results. While the supporting information includes a presentation of only 20 cycles I-V sweep, a more thorough investigation, e.g., overall cycle stability and detailed pulsed test results, is recommended.

Following the Reviewer’s request, a detailed characterization in quasi-static and pulsed regimes is presented in Fig. R1. Fig. R1a shows the quasistatic I-V curves for 8 devices with 100 cycles carried out with Agilent 4156C semiconductor parameter analyzer, with a compliance current between 1 μ A and 10 μ A to protect the devices from failure and possible permanent short circuit. The figure also shows an example of a representative I-V curve of the specific device. The device-to-device variability (D2D) is not significant for our purposes (please, see Supplementary Note 8 which shows the variability of switching probability after the discussion with Reviewer #2 point 3), with set voltages always between 0.2 V and 0.4 V with different distributions (see Fig. R1b). The hold voltage (considered as the voltage at which the current falls to a range below 10 nA) has a similar behavior. The cycle-to-cycle variability (C2C) shows broadened distributions for each specific device and the presence of a large variance is responsible for the switching probability properties used as a backbone of the presented work. Lastly, Fig. R1c shows the evolution of the I-V curves during the cycling, considering that an I-V curve has an overall duration of 3-5 seconds. Both the set and the hold voltages are stable within a range of values, without degradation toward higher or lower values. Some minor variations of the switching voltage can be attributed to the relatively immature RRAM technology in our clean room, which can be easily improved in an industrial process scale.

Fig. R1: Quasi-static characterization for different Ag-based volatile RRAM devices on the same chip. **a.** I-V curves for 8 different devices. **b.** Distributions of set voltage and hold voltage for 8 different devices. **c.** Evolution of set voltage and hold voltage during cycling.

The temporal dynamics of the RRAMs in the pulsed regime was then characterized by applying triangular pulses, from which the switching voltage can be monitored. Fig. R2a shows the voltage pulse and 100 current traces as a result of the stimulation. A 200 mV initial reading pulse is applied to check the device state, which was the high resistance state. (HRS) in all cases. Figure R2b shows the set voltage as a function of the measurement number along 10^5 cycles, indicating no significant deviation. Note that the set voltage is relatively large compared to the values reported in the text, due to the different time scales.

Fig. R2: Triangular pulse response of Ag-based volatile RRAM. **a.** Voltage pulse trace and corresponding current response for 100 different cycles. **b.** Set voltage as a function of the measurement cycle.

Fig. R3 shows the response of the device to a rectangular pulse for 100 cycles, again indicating substantial stability of the RRAM switching behavior.

Fig. R3: Current response to rectangular programming voltage pulse for 100 cycles of Ag-based RRAM.

To provide a comprehensive electrical characterization of our devices as suggested by the Reviewer, we included Fig. R1, R2, and R3 in Supplementary Note 3 (pp. 3-5).

3. The individual transfer property of the adopted transistor should be provided.

The transistor adopted in this work is based on a standard industrial CMOS fabrication process. Fig. R4 shows the transistor characteristics in linear and logarithmic scales and the transcharacteristics for different V_{DS} . We have now added Fig. R4 in Supplementary Note 2 (p. 3).

Fig. R4: Characterization of MOSFET **a.** Characteristic of the adopted MOSFET in linear y-scale. **B.** Characteristic of the adopted MOSFET in logarithmic y-scale. **C.** Transcharacteristic of the adopted MOSFET.

4. Several abbreviations lack of specific explanation when first appear.

We apologize for not having provided sufficient clarity in the first submitted version and thank the Referee for pointing this out. We revised the main text to give the appropriate explanation for all the abbreviations (p. 3).

5. In Figure 2, the authors applied 1~2 V voltage pulses to switch the device ON. However, in Figure 1c, it is evident that the necessary threshold voltage (V_{th}) for switching is less than 0.4 V. This prompts the question of why a larger pulse voltage is required for switching, especially considering the relatively low V_{th} indicated in the previous figure. Furthermore, the switching probability under 1 V bias appears to be small, even when subjected to high-frequency pulse stimulation. Clarification regarding this apparent discrepancy is necessary.

The difference in threshold voltages between the experiments shown in Figure 2 and the I-V curve in Figure 1c can be attributed to the different timescales of the experimental characterizations. In Figure 1c, the I-V curve of the device is a DC characteristic, obtained through a quasi-static measurement with an Agilent 4156C semiconductor parameter analyzer. In the experiments reported in Figure 2, the device is operated in the pulsed regime, where rectangular voltage pulses are applied with a duration $t_{pulse}=2.5 \mu s$ and variable amplitude and frequency of the train of pulses. In this regime, the duration of the application of a voltage across the device is much shorter compared to the DC regime characterization, resulting in a higher voltage required for the switching. This well-studied behavior in metal oxide-based RRAM is called the “*Voltage-Time dilemma*” which highlights an intrinsic tradeoff between the speed and voltage required in RRAM switching [1]. As a general rule, for an exponential decrease of pulse duration, a linear increase in pulse amplitude is needed to induce switching in the RRAM device [2]. This effect generally constrains the application of RRAM in some specific contexts such as storage class memory: if there is a need for high-speed programming of the memory (in the order of nanoseconds), one needs to apply higher voltages in the system, which also impacts the reliability of the device [3]. However, for neuromorphic systems, the requirements for speed are relaxed thus overcoming the issue of high voltages.

To illustrate the voltage-time dilemma on our device, we carried out the experiment reported in Fig. R5 for increasing pulse duration, namely 10 ms, 1 ms, 100 μs and 10 μs . As expected, the switching voltage increases with the shortening of the pulse duration.

Fig. R5: Device response for different programming pulse durations **a.** Voltage pulse trace and corresponding current response for different programming pulse durations. **b.** Set voltage as a function of programming pulse duration.

To better clarify the difference between characterization techniques and the effect of their different timescale, we modified Fig. S1 of the Supplementary material to include both DC characterization and pulsed regime characterization setup schemes (also reproduced in Fig. R6 for convenience) and we included Fig. R5 in the Supplementary Material. Moreover, we further emphasized the different characterization procedures in the main text in the “Stochastic switching of volatile RRAM devices” section (p. 4).

Fig. R6: New version of Fig. S1 of Supplementary Note 1 “Experimental setup and measurement procedure”.

References

1. Waser, Rainer, et al. "Redox-based resistive switching memories—nanoionic mechanisms, prospects, and challenges." *Advanced materials*, **21**, 2632-2663, (2009).
2. Huang, Peng, et al. "Analysis of the voltage–time dilemma of metal oxide-based RRAM and solution exploration of high speed and low voltage AC switching." *IEEE Transactions on Nanotechnology*, **13**, 1127-1132, (2014).
3. Menzel, Stephan, et al. "Physics of the switching kinetics in resistive memories." *Advanced functional materials*, **25**, 6306-6325, (2015).

6. The specific meaning of the y-axis in Figure 3c should be clarified to ensure a clear interpretation of the data. In addition, an explanation is required for the discrepancy between Figure 3b and Figure 3c, where none of the devices switch ON at a frequency of 20 Hz in Figure 3b, but a small probability of one device switching ON is observed in Figure 3c.

We thank the reviewer for raising this point. In Figure 3c, the y-axis represents the probability of observing a specific quantity of ON cells (shown on the x-axis) for a particular input pulse train frequency. Data in the histograms of Figure 3c are obtained from several experimental trials and, as pointed out by the Reviewer, a small probability is observed for $f = 20$ Hz. On the other hand, in Figure 3b for $f = 20$ Hz, we show a single trial of the experiment in which none of the cells switched on. We choose to show a single trace example in Figure 3b of the most probable configuration that can be determined from the histograms of Figure 3c also reported in the Tab. R1.

To better clarify the results in Fig. 3c, we modified the main text in the section "RRAM circuit for frequency sensing" (p. 5) and the figure description (p. 21), thus allowing a better interpretation of the data.

Frequency	20 Hz	112 Hz	632 Hz	3.5 kHz	20 kHz
The most probable number of ON cells	0	1	1	2	3

Tab. R1: The most probable number of ON cells for each frequency for the experiment of frequency sensing of Fig. 3 in the main text.

7. According to Kirchhoff's law, an increase in the number of ON-state devices typically results in a larger output current. However, a curious observation arises in Figure 3b, where the current levels for the cases of 1 device ON and 2 devices ON appear to be similar, except for the instance at $f = 20$ kHz. The authors should provide an in-depth analysis of the factors contributing to this peculiar behavior.

We agree with the Reviewer that an increase in the number of ON-state devices should result in a larger output current. In particular, in the 1T1R configuration, the device current I_C in the ON-state is controlled by the transistor, thus the total current is equal to:

$$I_{TOT} = N_{Devices,ON} * I_C.$$

This general trend is also observed in our experiment, e.g., Figure 3b shows that, for $f = 20$ kHz, there are three distinct current levels depending on the number of on-state devices. For $f = 3.5$ kHz, two current levels are present, which can be better noticed from the raw data trace (light blue curve) instead of the averaged data (dark blue curve) of the first submitted version. As also pointed out by Reviewer #2 (point 4), the data representation for Figure 3b was not sufficiently effective for which we apologize. The visualization problems in Figure 3b are related to the spike density that exponentially increases with the frequency. Fig. R7a shows the raw current trace of Figure 3b for $f = 3.5$ kHz with different timescales, which highlights the clarity issues.

Fig. R7: Plot of the raw current trace of Fig.3 for $f = 3.5$ kHz **a.** Different aspect ratios: when the temporal x-axis is large (left), it becomes challenging to discern the current level and the number of ON devices compared to a compressed version of the same plot (right). Moreover, considering that the pulse duration is $2.5 \mu s$, it is difficult with a time scale of tens of milliseconds to tell capacitive spurious spikes from device switching. **b.** Four closer looks at individual spikes within the raw current trace, focusing on a $40 \mu s$ time window: spurious capacitive spikes involve just the first point of the programming pulse when present.

Moreover, being the pulse duration equal to $2.5 \mu s$, is difficult to recognize the device switching from spurious capacitive spikes by watching the raw traces in the time scale of tens of milliseconds: they can be visually spotted just thanks to a temporal scale zoom as showed in Fig. R7b. These spurious spikes are due to a capacitive-coupled disturbance referable to the experimental setup (coaxial cables, manipulators, chuck, instrumentation, etc.). It is possible to

readily eliminate the spurious spikes through a simple low pass filter (LPF) thereby enabling accurate extraction of the device conductance (Figure 2b) and of the device state (Figure 3c). It is important to note that this disturbance effect should not arise in a fully integrated version of the chip, where parasitic capacitance can be eliminated by careful layout design.

To allow a better visualization of the current traces according to the Reviewers' suggestions, we modified Fig. 3 to report just the smoothed version of the temporal current traces in the main text. In this new version, we used an LPF with a smaller time window (mobile median filter window = 7 samples) to prevent any drop in the amplitude of the signal as in the first submitted version (mobile mean filter window = 15 samples), where it was not possible to properly recognize the number of ON devices in the case of $f=3.5$ kHz. We moved the raw traces in Supplementary Note 9 where we also discuss the presence of spurious capacitive spikes including the plots of Fig. R7 (p. 9).

Fig. R9: The new version of the Fig.3 in the main text.

8. When considering the actual situation, I recommend using the term 'specific' rather than 'average' to describe the number of devices in the ON state as a function of the input train frequency in Figure 3d.

In Figure 3d, the number of ON devices (y-axis) is calculated through the average on the ensemble of experiments on the circuit for frequency sensing. We hope that this can clarify our choice to use the word “average” instead of “specific” in the description of Figure 3d.

Reviewer #2:

The authors present innovative neuromorphic circuits for spatio-temporal signal processing using silver-based volatile ReRAM. They successfully emulate the audio signal processing system in the human cochlea, achieving the tonotopic mapping of temporal information and classification of audio signals based on their frequencies. The work is commendably innovative and well-supported with data. However, it is regrettable that the speech recognition aspect relies on simulation rather than real memristor crossbars. Several concerns should be addressed before fully endorsing this work.

We thank the Referee for the positive comments on our work and for the valuable suggestions to improve the manuscript. The goal of our speech recognition simulations is to show the capability of novel neuromorphic hardware primitives achievable with our memristive tonotopic map, while the classification part through a real crossbar is a technological challenge that we cannot face at the moment with our resources. Our responses to the comments are as follows.

1. Clarification is needed regarding the definition of "Pswitch" in line 13 on page 4. The numerator in the equation appears to be Ntrain minus Ntrain/on. This adjustment would render the observed trend of increasing Pswitch with higher applied voltage more reasonable.

We thank the Reviewer for raising this point. We defined the switching probability P_{switch} as the probability to switch on the device by applying a train of voltage pulses with amplitude V , frequency f , and duration $T_{window} = 25$ ms. To experimentally characterize P_{switch} , we apply N_{trains} to the device and count how many times the device switches on, namely $N_{trains|ON}$. To better clarify this point, we can write:

$$P_{switch} = \frac{N \text{ times event occurs}}{N \text{ trials}} = \frac{N_{trains|ON}}{N_{trains}}$$

where the event is "the device switches on". We perform the experiment of applying a single train of pulses with voltage amplitude V and frequency f to the device for N_{trials} which is equivalent to applying N_{trains} pulses to the device. Within these N_{trials} pulses, we count how many times the event has occurred i.e. how many times the device switched on i.e. $N_{trains|ON}$.

To summarize this definition:

- P_{switch} is the probability of switching on the device by applying a single train of voltage pulses with amplitude V , frequency f , and duration $T_{\text{window}} = 25$ ms. ($0 \leq P_{\text{switch}} \leq 1$)
- N_{trains} is the total number of trials of the experiment i.e. the number of trains of pulses applied to the devices with a given voltage amplitude V and frequency f .
- $N_{\text{trains}|_{\text{ON}}}$ is the number of trains that cause the device to switch on. ($N_{\text{trains}|_{\text{ON}}} \leq N_{\text{trains}}$)

Note that, based on the above definitions, we have $N_{\text{trains}} = N_{\text{trains}|_{\text{ON}}} + N_{\text{trains}|_{\text{OFF}}}$ being the state of the device either ON or OFF.

To improve the clarity of this definition, we modified the main text in the section “Stochastic switching of volatile RRAM devices” (pp. 4-5).

2. Figure 2c should provide the pulse waveform to validate that the waiting time between each spike remains lower than the retention time.

The current response of Figure 2c is related to the voltage pulse waveform of Figure 2a for the temporal features but the voltage amplitude of the applied train is different being $V=1.5\text{V}$ for the first subplot, $V=1.75\text{V}$ for the second, and $V=2\text{V}$ for the third. As requested by the Reviewer, we provide in Fig. R10 the pulse waveform associated with every current trace of Figure 2c. The waiting time between each spike in the Figure is equal to $t_{\text{period}} = 285 \mu\text{s}$ which is much lower than the retention time of these devices in the order of 50-100 ms (please refer to Supplementary Note 5 for retention time distribution).

Fig. R10: a. Applied voltage pulse waveforms and relative current responses for different voltage amplitudes. b. Parameters of the voltage pulse waveform. c. Zoom of the voltage pulse waveform.

To address the point highlighted by the Reviewer we included Fig. R10 in the Supplementary material (p. 7).

3. It is advisable to include the variation of P switch in Figure 2 to facilitate comprehensive probabilistic studies. The impact of the variance observed in these measurements is noteworthy, and given its inherent presence, addressing this factor is essential.

We conducted a detailed analysis of the switching probability in our devices to understand their behavior and how to manage their properties for specific applications. To perform the presented measurement, we took all the necessary precautions for this type of probabilistic study, i.e., we set the train amplitude and frequency randomly to each cycle to avoid correlation effects. Subsequently, in the next cycle of the measurement, we randomly changed both the voltage and frequency of the train again. We tested every possible combination of selected voltage and frequency for the desired number of cycles. Fig. R11 illustrates this measurement procedure.

$$f = [20 \text{ Hz} , 112 \text{ Hz} , 632 \text{ Hz} , 3.5 \text{ kHz} , 20 \text{ kHz}]$$

$$v = [1 \text{ V} , 1.25 \text{ V} , 1.5 \text{ V} , 1.75 \text{ V} , 2 \text{ V}]$$

Matrix of combinations (25 x 2)	$M_C = \begin{bmatrix} 1 \text{ V} & 20 \text{ Hz} \\ 1 \text{ V} & 112 \text{ Hz} \\ \dots & \dots \\ 2 \text{ V} & 3.5 \text{ kHz} \\ 2 \text{ V} & 20 \text{ kHz} \end{bmatrix}$
Randomized matrix of combinations (25 x 2)	$M_{C, \text{rand}} = \begin{bmatrix} 1.25 \text{ V} & 632 \text{ Hz} \\ 1 \text{ V} & 20 \text{ Hz} \\ \dots & \dots \\ 2 \text{ V} & 632 \text{ kHz} \\ 1 \text{ V} & 112 \text{ kHz} \end{bmatrix} \leftarrow i$

Fig. R11: Measurement protocol for switching probability characterization to avoid correlation effects.

To provide a proper answer to the Reviewer, we performed extensive characterization of our devices, reported in Fig. R12. Here, we show the switching probability matrices for 8 different devices with 2500 trains (5 voltage amplitudes x 5 frequencies x 100 cycles) and more than 300k spikes corresponding to 6 hours of characterization for each matrix:

Fig. R12: Switching probability matrices for 8 different devices.

It is worth noting that there exists a natural device-to-device variability in switching probability due to statistical variability of the threshold voltage (please refer to the new extended electrical characterization in Supplementary Note 3). However, this variability does not hinder the objectives of our work. In fact, all the matrices of Fig. R12 fall within the same operational range of voltages and frequencies. This alignment is sufficient for our research focus, as also demonstrated by the frequency sensing experiment presented in Figure 3 of the main text, which involves three different devices. The device variance mentioned by the Reviewer is, in fact, an integral component of our work on switching probability, and it was accounted for in all the simulations discussed in our manuscript.

To further address this point, we carried out new measurements with a higher number of cycles, focusing on the variability of switching probability captured by the parameter $\Theta = V(P_{\text{switch}}=0.5)$ i.e. the voltage amplitude required to achieve a switching probability of 50%. In fact, as explained in the main text and by our probabilistic model, the variability of set voltage determines the switching probability curve, and so the variability of switching probability can be captured by the variability of characteristic voltage Θ . As reported in Fig. R13a, we found that this value also exhibits a device-to-device variability, with a standard deviation equal to $\sigma_v = 0.054$, which is consistent with the value of $\sigma_v = 0.05$ previously found and used in our simulations. Furthermore, Fig. R13b also illustrates the impact of increased variability on the frequency sensing experiment. The primary consequence of increasing variability is its effect on the perturbation of the linearity

of the characteristic of frequency sensing. However, this effect remains limited also in the case of a large value of $\sigma_v = 0.2$.

Fig. R13: a. Switching probability curve for 10 different devices ($f=632$ Hz): there is a natural variability of the curve that can be captured through the characteristic voltage $\Theta = V(P_{\text{switch}}=0.5)$ that has a standard deviation equal to $\sigma_v = 0.054$. b. Impact of the variability of switching probability on frequency sensing circuit of Fig.3: Simulation agrees with experimental results as shown in Fig. S14. If the variability is higher, the linearity of the curve starts to be impacted, still well performing for frequency sensing on a logarithmic scale.

To conclude, the switching probability is also stable in time after the extensive use of the device as reported in Fig. R14, where we show the switching probability matrix of a device after 1 million pulses (and intermediates points) to track its evolution.

Fig. R14: Stability of switching probability in time after prolonged use of the device (1M pulses).

To provide a comprehensive study on the variability of switching probability as suggested by the Reviewer, we included Fig. R12, 13, and 14 in the dedicated note “Switching probability variability” of the Supplementary Material (p. 8). Moreover, we included Fig. R11 in the Supplementary

Material, and we also added the new section “Switching probability measurement protocol” to the “Methods” in the main text (p. 13).

4. In the experimental demonstration of the RRAM circuit for frequency sensing (Figure 3b), it's challenging to discern the number of devices switched on for temporal traces with frequencies exceeding 3.5kHz. A more discernible plotting method should be considered.

We agree with the Reviewer that in the first submitted version of Fig. 3b, it is difficult to see the individual pulses of current for $f = 20$ kHz due to an exponential increase of temporal spike density i.e. 20 spikes in 1 ms. Nevertheless, despite the spikes being compressed on the temporal x-axis, the number of ON devices can be derived from different current levels on the y-axis, both on raw and smoother current traces that show current steps. However, we have reached the conclusion that plotting both raw and smoothed traces in the same plot is misleading and does not add any useful information.

As suggested by the Reviewer, we modified the plot of Fig. 3 to facilitate the visualization of the current levels by just reporting the smoothed traces with a smaller averaging window for the low pass filter (mobile median filter window = 7 samples) to prevent any distortion of the amplitude of the smoothed current trace as in the previous version, where it was difficult to spot different current levels (mobile mean filter window = 15 samples). The raw current traces were moved from in Supplementary Note 9 (p. 9). For convenient reference we also report the new version of Fig.3 of the main text in Fig. R16.

Fig. R16: New version of the Fig.3 in the main text.

5. In the section of speech recognition (figure 5c), three XOR outputs may be not sufficient for extracting audio features, more than 5 outputs are recommended.

We thank the Reviewer for the suggestion to improve the capability of our system. Our choice of three XOR outputs was made considering that the number N_{OUT} of XOR outputs is linked to the number of memristive devices for each memristive tonotopic map (MTM) needed for the cochlear processing and so to the number of required channels in the analog-to-spike conversion. Moreover, N_{OUT} also determines the number of weights of the cortical network for interpretation being the number of weights of the first layer equal to $N_{W1} = N_{OUT} \times 64$ as illustrated in Fig. R17. We have kept N_{OUT} as low as possible to showcase the capabilities of our system with a minimal number of elements, thus gaining in power consumption and occupied area, while still ensuring excellent accuracy in classification of 96.5%. This value, in fact, is in the order of best automatic speech recognition systems (ASR) implemented with deep learning (DL) techniques at the edge [1]. Note that a direct comparison with standard speech recognition algorithms is out of scope, since, e.g., we have used a limited set of words compared to DL-ASR. Anyway, assuming a threshold for accuracy in ASR of around 95%, our choice of $N_{OUT} = 3$ can be considered a suitable value to reach sufficient accuracy.

In line with the Reviewer's suggestion, we found that an increase in the number of XOR outputs to $N_{OUT} = 6$ leads to an increase in accuracy up to 100 % on the test set, at the expense of 3 additional XOR gates, 3 additional memristive devices for each MTM, and an increased number of synaptic weights for the first layer of the cortical neural network given by:

$$\Delta N_{W1} = N_{W1, N_{OUT}=6} - N_{W1, N_{OUT}=3} = 6 \times 64 - 3 \times 64 = 192$$

which corresponds to double the number of weights compared to the previous case with $N_{OUT} = 3$. In this case, we can save resources by decreasing the number of neurons in the cortical network. Decreasing the number of neurons to $N_{neurons}=16$, the accuracy drops to 97.5%, which is still above the reference 95%. To show this trade-off between N_{OUT} and $N_{neurons}$, we analyzed different configurations reported in Fig. R17, where we show the example confusion matrix for a test for each configuration.

Fig. R17: Analysis of the impact on the accuracy of the number of XORs (N_{OUT}) and the number of neurons of the cortical network ($N_{neurons}$).

To conclude, the Reviewer's suggestion is a crucial point for the up-scaling of the system: a larger set of spoken words would certainly benefit from a higher N_{OUT} . However, the goal of this work is to demonstrate a novel approach for audio processing with volatile memristive devices. In this regard, the speech recognition application supports the interpretability of the proposed memristive tonotopic map by a neural network in analogy with the processing of the auditory cortex after the biological tonotopic cochlear mapping [2]. The scale-up of the overall computing system goes beyond the scope of this work.

We have incorporated this discussion and Fig. R17 within the Supplementary Material (please refer to Supplementary Note 13, p. 13) to enhance the comprehensiveness of our study and we modified the “Interpretability of tonotopic map: speech recognition” section of the main text to address this crucial point for the up-scaling of the system (p. 8).

References

1. Peinl, R., Rizk, B., & Szabad, R., “Open source speech recognition on edge devices”, 10th International Conference on Advanced Computer Information Technologies (ACIT), IEEE, (2020).
2. Kandel, E. R., Schwartz, J. H. Jessell, T. M., Siegelbaum, S., Hudspeth, A. J., & Mack S. (Eds.), Principles of neural science, McGraw-Hill, 4, 1227-1246, (2020).

6. In the section of speech recognition (figure 5), only frequency signals are considered while the amplitude of audio signals is disregarded. the duration of distinct frequencies significantly influences the output of the XOR results. This temporal aspect constitutes another crucial factor that necessitates consideration.

For the application to speech recognition, the information about the amplitude of audio signals is captured in the analog-to-spike conversion thanks to 3 different thresholds used to generate 3 different pulse waveforms that are fed to the memristive tonotopic map circuit (please, refer to Supplementary note 10 for the block diagram of analog-to-spike conversion, also reproduced in Fig. R18 in a revised version for convenient reference).

Fig. R18: Conversion from audio temporal signal to spike trains.

We agree with the Reviewer that the duration of different audio signals influences the output of XOR results, which is a desired positive effect for our system. A single word contains different phonemes which in turn contain different frequency components and different sound wave

pressures but also different durations [1]. Similar to the primary auditory cortex of the brain, the time duration of the audio signal is a feature for classification itself [2]. This is a typical approach used in neuromorphic systems that have to deal with spatiotemporal signals, where time is the domain for the computing task, while the temporal characteristics of the signals are the elements involved in computation [3]. Moreover, in an asynchronous approach for the use of our hardware primitives, the duration of the signal is itself the trigger for the computing task: the starting point of the signal initiates the computation while the end of the signal induces the system to relax to the resting state, without any a-priori knowledge of the arrival time or duration of the audio signal.

Following the point raised by the Reviewer, we emphasized this point in the “Interpretability of tonotopic map: speech recognition” section of the main text (p. 8) and in Supplementary Note 12 (p. 12) and we added the “Analog-to-Spike conversion” section in “Methods” of the main text (p. 14).

References

1. Lord, H. W., Gatley, W. S., & Evensen, H. A., *Noise control for engineers*, McGraw-Hill, (1980).
2. Mesgarani, N., David, S. V., Fritz, J. B., & Shamma, S. A., *Phoneme representation and classification in primary auditory cortex*, *The Journal of the Acoustical Society of America*, **123**, 899-909, (2008).
3. Roy, Kaushik, Akhilesh Jaiswal, and Priyadarshini Panda. "Towards spike-based machine intelligence with neuromorphic computing.", *Nature*, **575**, 607-617, (2019).

Overall, the author's contribution is significant, though the concerns outlined above deserve attention to augment the credibility of their findings.

Reviewer #3:

The authors describe a RRAM-based circuit that could operate as an audio signal processing system. The RRAM devices used in this work are two terminal Ag/HfO_x memristors. Probabilistic model of the RRAM is developed based on measurements of device switching probability versus pulse frequency or voltage amplitude. The manuscript discusses several advantages of this system, such as the unique capability in interpreting audio frequency signals and doing tonotopic mapping.

I believe the performance and capabilities of the presented system are impressive, even though I have some comments/questions that should be addressed.

We thank the Referee for the positive comments on our work and for the valuable suggestions to improve the manuscript. Our responses to the comments are as follows.

1. Novelty at the system level is obvious, which is about the unique audio signal processing functionality. However, novelty at the device level seems not clear to me. First, the memristor structure and device mechanism in this work (Fig. 1) is nothing different compared to most other well-established memristors. Second, the switching probability of a memristive device versus pulsing frequency and voltage amplitude have both been studied in previous literature [such as *Nanoscale* 5, 5872–5878 (2013); *Nat. Commun.* 12, 5710, (2021)]. The trend of switching probability provided in this work (Figs. 2-3) is thus well expected. The author should pay some attention in explaining about their novelty at the device level, otherwise the need of having Figs 1-3 for discussing what they found about the RRAM device is undermined.

First, we want to clarify the novelty of our memristive devices. In our work, the RRAM device is volatile, i.e., there is no need for a reset pulse to restore the low resistive state, since the conductive filament spontaneously dissolves to minimize the surface energy after a retention time [1]. The I-V curve in Figure 1c shows this behavior, namely, the device is in the low resistive state after switching, then, when the applied voltage drops below the holding voltage V_{hold} , the device switches back to the high resistive state. Since the device serves as the foundation for all subsequent applications, it is worth noting that the I-V curve depicted in Figure 1c also serves as a point of reference for general device characteristics. These characteristics include a low set voltage, low current switching, and unipolar switching, which distinguishes our device from other memristive devices. Focusing again on the device side, previous works on switching probability

(including the ones mentioned by the Reviewer) rely on devices that need a reset procedure to come back to the high resistive state via bipolar voltage pulses [2-4]. Moreover, volatility is an important property for the spontaneous relaxation of the system to a well-defined resting state as also discussed in the next question of this discussion.

We also want to emphasize novelty of the experimental characterization and methodology of the switching probability in Figure 2. In our work, we do not define the switching probability on a single pulse, rather the switching probability is generalized and defined with respect to a train of pulses. Here, the switching probability is the probability for the device to switch on by applying a train of pulses with voltage amplitude V and frequency f in a time window T_{window} , thus introducing a conceptual difference with respect to previous works [2-4]. Such a general methodology is more useful in the context of neuromorphic systems that deal with trains of pulses instead of single pulses in analogy with their biological counterpart [5]. Moreover, since all memristive devices are vulnerable to intra- and inter-device variations, we believe that this approach is a better alternative to improve the robustness of memristive neuromorphic systems as the same mechanism works for biological synapses [5-6].

Lastly, in our experimental characterization, we examine the influence of voltage amplitude and frequency within a broad parameter range. In particular, for frequency sensitivity, for the first time, we assess the characterization of a memristor on a logarithmic scale spanning three orders of magnitude as reported in Fig. 2. This approach goes significantly beyond previous studies, which primarily explored non-volatile switching probability using single-pulse stimuli on a linear scale. This distinction is of paramount importance, particularly in the context of implementing bioinspired systems like the auditory system, where signals span a logarithmic frequency range from 20 Hz to 20 kHz. This characterization constitutes the crucial feature for tonotopic mapping and all its applications. In fact, the characterization results demonstrate the capability of our devices for stochastic logarithmic integration, while previous studies on integration at the device level worked on limited linear saturable scales. Accordingly, in Fig. 3, we conducted a unique targeted experiment on the circuit for frequency sensing that represents the core for the computation in the reported applications.

Lastly, we believe that the combination of our switching probability characterization methodology, coupled with the verified probabilistic model, offers a novel powerful tool for characterizing and simulating the application of stochasticity in neuromorphic systems. This approach allows to

expand the space of exploration of a single device to logarithmic scales of frequency, thus constituting the backbone for developing more advanced applications in temporal signal recognition.

To better highlight the novelty of our work, encompassing our devices (Fig. 1), the characterization methodology and switching probability results (Fig. 2), and the experiments on logarithmic frequency sensing (Fig. 3), we have revised the main text of the "Stochastic switching of volatile RRAM devices" section (pp. 3-5), "RRAM circuit for frequency sensing" section (p. 6), and the "Discussion" section (pp. 10-12).

References

1. Wang, W. et al., *Switching dynamics of Ag-based filamentary volatile resistive switching devices—Part II: Mechanism and modeling*, *IEEE Transactions on Electron Devices*, **68**, 4342-4349, (2021).
2. Gaba, Siddharth, et al. "Stochastic memristive devices for computing and neuromorphic applications.", *Nanoscale*, **5**, 5872-5878, (2013).
3. Yan, Xiaodong, et al. "Reconfigurable Stochastic neurons based on tin oxide/MoS2 hetero-memristors for simulated annealing and the Boltzmann machine." *Nature Communications*, **12**, 5710, (2021)
4. Zahari, Finn, et al. "Analogue pattern recognition with stochastic switching binary CMOS-integrated memristive devices.", *Scientific reports*, **10**, 14450, (2020).
5. Lisman, J. E., *Bursts as a unit of neural information: making unreliable synapses reliable*, *Trends in neurosciences*, **20**, 38-43, (1997).
6. Branco, T. & Staras, K., *The probability of neurotransmitter release: variability and feedback control at single synapses*, *Nature Reviews Neuroscience*, **10**, 373–83, (2019).

2. The transistor in the 1T1R structure keeps a RRAM being in a volatile state instead of a non-volatile state, eg. the CF spontaneously dissolves after a suitable retention time. How important is the this "volatile" property in realizing the following audio signal processing applications?

We have observed that, in the case of our device, volatility arises primarily from the materials properties and the fabrication process rather than being determined by the presence of the transistor [1]. This is supported by the characterization of the same device fabricated in 1R structure: in this case, the structure is a metal-insulator-metal (MIM) where the oxide layer is sandwiched between a platinum bottom electrode (BE) and a silver top electrode (TE). The fabrication starts with the electrodes which are patterned through 235nm-UV lithography and then platinum is deposited with the e-beam evaporation. A spacer layer of silicon dioxide is CVD-grown to separate the TEs from the BEs and finally, the active oxide and the silver metal electrodes are first patterned (as for the BE electrodes) and then deposited through the e-beam

evaporation without breaking the vacuum. Also in this case, the devices show a volatile behavior of resistive state as reported in Fig. R19a. To protect the device from permanent short-circuit and damage, an external resistor with $68\text{ k}\Omega$ resistance was used to limit the current. Moreover, for the 1T1R structure, we observed that the device is volatile also for different current compliance as shown in Fig. R19b.

Fig. R19: Quasi-static I-V curves of volatile RRAM for different structures. **a.** 1R structure. **b.** 1T1R structure.

The importance of the volatility property can be explained by the analogy with hair cells in the animal cochlea: In this context, sound waves propagate within the cochlear fluid, exerting mechanical stimulation upon the hair cells by applying a pressure capable of inducing their deformation. This mechanical deflection provides the trigger for the processing of the audio signals [2]. When the pressure is removed, i.e. when the sound is not present, the hair cells spontaneously come back to their resting position without the need for a control signal that prepares them for a new processing [3]. From a computational perspective, this mechanism demonstrates remarkable efficiency due to the presence of a single ground state in which the system can readily relax. This ground state serves as the initial condition for subsequent computations. In the same way, our system can spontaneously relax to an a-priori known state, i.e. all the devices in the OFF state after computation, thanks to the volatile memory of our devices. The spontaneous relaxation to this ground state simplifies the system by avoiding the need for a trigger signal and moving the paradigm to an asynchronous system. Furthermore, in contrast to other memristive devices, the volatility property eliminates the necessity for an electrical reset phase in the device and the need

for bipolar voltages. As a consequence, this leads to substantial energy and system complexity savings.

To emphasize the significance of volatility in light of the points raised in this discussion, we have further elaborated on this point in the "Stochastic switching of volatile RRAM devices" section (p. 3) and the "Discussion" section (pp. 11-12). Lastly, we included Fig. R19 in the Supplementary Note 15 (p. 15).

References

1. Wang, Wei, et al. "Switching dynamics of Ag-based filamentary volatile resistive switching devices - Part II: Mechanism and modeling.", *IEEE Transactions on Electron Devices*, **68**, 4342-4349, (2021).
2. Kandel, E. R., Schwartz, J. H. Jessell, T. M., Siegelbaum, S., Hudspeth, A. J., & Mack S. (Eds.), *Principles of neural science*, McGraw-Hill, **4**, 1227-1246, (2020).
3. Hudspeth, A. J., and Peter G. Gillespie. "Pulling springs to tune transduction: adaptation by hair cells." *Neuron*, **12**, 1-9, (1994).

3. Clearly the raw experimental data in the middle panels in Fig. 3b shows occasional spikes could be at high current level (~20-40 uA) due to the stochastic device behavior. If one high spike appeared, how could the authors determine if only one device is ON at this moment instead of interpreting that two devices are ON? It seems interpreting over single event is less reliable than interpreting the statistics over ensemble of multiple events. Will utilizing the stochastic property of RRAM in hardware require more energy due to the need to collect ensemble behaviors of a RRAM, as opposed to relying on a single event of a RRAM? The energy consumption could scale up quickly and become problematic if the number of RRAM devices increases.

The occasional spikes in the raw experimental data in Fig. 3b are due to the presence of a capacitively-coupled disturbance due to the ultimate limitation of the experimental setup (coaxial cables, manipulators, needles, chuck, instrumentation) rather than the stochasticity of the device. Given the time scale of Fig.3b in the order of tens of milliseconds and the programming pulse equal to $t_{\text{pulse}} = 2.5 \mu\text{s}$, we replotted the data of Fig. 3b on a shorter time scale to properly discriminate spurious spikes from device switching in Fig. R20. The spurious spike systematically occurs only at the initial point of the programming voltage pulse, owing to a rapid discharge process that is significantly shorter in duration compared to the programming pulse itself ($t_{\text{pulse}} = 2.5 \mu\text{s}$). Spurious spikes can be readily eliminated with a low pass filter (LPF), thereby enabling

accurate extraction of the device conductance as demonstrated in Figure 2b and of the device state as reported in the histograms of Figure 3c. It is important to note that this disturbance effect would not arise in an integrated version of the chip, eliminating the coupling sources for the disturbance.

Fig. R20: a. Raw current trace of Fig. 3b for $f = 3.5 \text{ kHz}$. b. Zoom of raw current traces for four different regions where it is possible to spot spurious capacitive spikes from device switching.

From this analysis it is clear that high-current spikes are just spurious, and they can be easily removed, going in the direction of making the single RRAM device more reliable for the identification of the real number of ON devices based on the current level. However, in general, we agree with the Reviewer that interpreting computation on single events is less reliable than interpreting the statistics over the ensemble of multiple events. Unlike conventional analog neuromorphic systems [1], our work does not aim to mitigate stochasticity, rather to exploit it to obtain unique functionalities such as logarithmic integration. Additionally, the concept of “multiple events” is necessary in neuromorphic systems where information is not carried out in a single event [2]. This concept is intrinsically included in our novel definition of switching probability, which relies on a train of pulses instead of a single pulse i.e. on multiple events instead of a single event. As pointed out by the Reviewer, increasing the number of RRAM devices the energy consumption increases. In this work, the compliance current (around $I_c = 16 \mu\text{A}$) is set just to be properly readable with the oscilloscope. However, as demonstrated in the I-V curve reported in Fig. R19b, the device can switch with a 10 nA current thus reducing the energy consumption by more than 1000 times compared to our data in Fig. 3. This provides enough margin to increase the number of RRAM devices without affecting energy consumption and still requiring a reasonably small area (Please refer to the next point of this discussion for the comparisons with CMOS solutions). Lastly, our system enables logarithmic integration using a probability mechanism, thus

removing the need for energy-hungry and large capacitances for integration operations. As a result, the device reduces the number of spikes, thus contributing to overall energy and area efficiency.

In conclusion, regarding Fig. 3, and also according to other Reviewers' comments, we realized that combining raw and smoothed traces in the same plot is misleading. We have revised Fig. 3 to show only smoothed traces with a different low-pass filter on a smaller number of points to avoid excessive impact on the current amplitude. Raw traces are now presented in Supplementary Note 9 (p. 9) with a discussion on spurious spikes. Additionally, we addressed energy consumption scaling in the new version of the "Discussion" section in the main text (p. 11).

References

1. Payvand, Melika, et al. "A neuromorphic systems approach to in-memory computing with non-ideal memristive devices: From mitigation to exploitation." *Faraday Discussions* 213 (2019): 487-510.
2. Liu, Shih-Chii, et al., eds., "Event-based neuromorphic systems", John Wiley & Sons, 2014.

4. Implied in the discussion is that the motivation for using memristive devices to replace CMOS technology system-level approach in mimicking biological functionalities are mainly due to advantages in power consumption, small area occupation, unique functionalities and so on. I must ask where in this paper is evidence that this RRAM device does not suffer from the same challenges? In other words, the author should provide some direct comparison or benchmarking of power consumption and circuit footprint with CMOS technology and make some strong argument how RRAM in this work is advantageous over CMOS to better support their discussion.

The point raised by the Reviewer deserves examination from two distinct perspectives, namely the computational and the technological sides.

From the computational side, we investigate a "device-level approach" for neuromorphic audio processing. This approach aims to maximize the utilization of the device inherent physics and dynamic properties for computation. It involves a shift from the conventional practice of *simulation* of biological neural networks typically employed in CMOS-based neuromorphic systems, to the *emulation* of neural and physiological mechanisms [7], as schematically depicted in Fig. R21. The usage of volatile memristive devices in this work provides unique functionalities: volatile RRAM offers complex dynamics thanks to internal physics mechanisms that directly resemble the emulation of biological mechanisms. This is not straightforward with CMOS electronics, since it would require a circuit rather than a single device to simulate a particular

neural behavior. This transition represents a paradigm shift rather than a simple technology shift, therefore drawing a comparison with CMOS systems is rather challenging. Implementing neural primitives directly in hardware using a single device simplifies the system and aligns with the goals of making neuromorphic computing more explainable and biologically plausible [8]. This is a crucial aspect when scaling up the size and capabilities of neuromorphic computing systems, where system complexity can have a significant impact [8,9]. Moreover, explainability and biological plausibility are fundamental requirements to integrate the powerful computational models developed by the neuroscience community in hardware [9,10].

Fig. R21: Computational approaches of Neuromorphic computing: Including memristors into the computing systems move the paradigm from a simulation of biological neural network to the emulation of their functionalities.

From a technological standpoint, one of the unique features of volatile RRAM is its ability to internally perform stochastic logarithmic integration. On the other hand, for a CMOS-based system to achieve even a basic linear integration function through a leaky integrate-and-fire (LIF) neuron, one would require a substantial capacitance to handle typical audio time constants, which can be as long as 50 ms. This requirement would result in a relatively-large physical footprint and a specific design approach, as illustrated in Fig. R22a where the area occupation of different CMOS LIF neuron implementations is reported. In fact, the size of the capacitance in CMOS systems increases proportionally with the time constants, while in our system the time constant values are decoupled from the area occupation.

Additionally, our systems work at biological frequencies, that are low compared to most CMOS neuron implementations (in the order of MHz), directly gaining advance on power consumption as

evident in Fig. R22b. At the same time, we also use short pulses compared to the time scales of audio signals, because in our system it is possible to decouple the spike duration ($t_{\text{pulse}} = 2.5 \mu\text{s}$) from the operative time constants (up to 50 ms). In this way, we can reach an energy consumption per spike in the order of 40 pJ as reported in Fig. R22c. This value is not a real benchmark due to the fact that the compliance for the experiment is fixed to a reasonable value that can be read through the oscilloscope. The compliance current value can decrease up to 10 nA thus reducing the energy consumption per spike to a value of 40 fJ.

Fig. R22: Comparison between parameters of a single CMOS leaky integrate-and-fire (LIF) neuron implementation from different works in 65-nm technology and this work. Due to their interesting results, we also included a work that couple CMOS with nano electromechanical systems (NEMS) [6] and a work using pseudoresistors [4] **a.** Area occupation versus characteristic time constant. **b.** Power consumption versus characteristic time constant. **c.** Energy per spike versus characteristic time constant.

The area occupation of this work is estimated to be the worst case for RRAM technology referring to $100F^2$ [11]. The energy of the single spike is calculated as:

$$E = V_{in} I_C t_{pulse}$$

The upper limit for the power consumption is calculated by assuming that all the pulses in the train cause a switch, namely:

$$P_{lim} = \frac{N_{pulses}(\tau) V_{in} I_C t_{pulse}}{T_{window}} = \frac{T_{window}}{\tau} \frac{V_{in} I_C t_{pulse}}{T_{window}} = \frac{E}{\tau}$$

While the real power consumption is calculated by extracting the average number of ON spikes in the experiments for the intermediate voltage amplitude of 1.5V:

$$P = \frac{N_{pulses}(\tau)V_{in}I_C t_{pulse}}{T_{window}}$$

Last but not least, our devices are volatile, i.e., they naturally return to a well-defined resting state without the need for a reset procedure that is power consuming and increases the circuit complexity. Moreover, our devices can be asynchronously triggered again when a new signal is detected, thus eliminating the need for constant control signals as in synchronous CMOS systems.

Although not yet optimized to achieve the best performance in energy and area occupation, our approach constitutes a proof-of-concept for a novel computing hardware paradigm. At the same time, there is a huge margin for the optimization of energy consumption and area occupation thanks to the strong scaling capability of RRAM technology and the paradigm shift.

To provide a comparison with CMOS standard technology we included the paradigm shift scheme (Fig. R21) and benchmarks (Fig. R22) in Supplementary Note 15 (p. 15) and we included the point here discussed in the new “Discussion” section in the main text (pp. 9-12).

Tab. R2 summarizes the points discussed about this work versus CMOS implementations.

Parameter	This work	CMOS based systems
Power Consumption	[x] Lower operative frequency [x] Short spikes [x] No need for a reset procedure [x] Need for parallelization	[x] High spiking frequency [x] Integration based on charging a capacitance
Area occupation	[x] No need for a large capacitor [x] Primitive implemented with a single device [x] Need for 1T1R structure	[x] Need for large capacitor [x] Primitive implemented with a circuit
Biological plausibility and explainability	[x] Biological emulation of hairy cells and auditory cortex [x] Biological operative frequencies	[x] Simulation of mathematical equations
Complexity	[x] No need for Reset [x] Device-level emulation (no need for the design of biological mechanism) [x] Immature technology [x] Asynchronous	[x] Circuital simulation (need for the design of biological mechanism) [x] Mature technology

Tab R2: Comparison of parameters between this work and previous CMOS neuron implementations.

References

1. Aamir, Syed Ahmed, et al. "A highly tunable 65-nm CMOS LIF neuron for a large scale neuromorphic system." *ESSCIRC Conference 2016: 42nd European Solid-State Circuits Conference*. IEEE, 2016.
2. Rozenberg, M. J., O. Schneegans, and P. Stolar. "An ultra-compact leaky-integrate-and-fire model for building spiking neural networks." *Scientific reports* 9.1 (2019): 11123.
3. Velichko, Andrei, and Petr Boriskov. "Concept of LIF neuron circuit for rate coding in spike neural networks." *IEEE Transactions on Circuits and Systems II: Express Briefs* 67.12 (2020): 3477-3481

4. Chen, Xiangyu, et al. "An ultra-compact leaky integrate-and-fire neuron with long and tunable time constant utilizing pseudo resistors for spiking neural networks." *Japanese Journal of Applied Physics* 61.SC (2022): SC1051.
5. Joubert, Antoine, et al. "Hardware spiking neurons design: Analog or digital?." *The 2012 International Joint Conference on Neural Networks (IJCNN)*. IEEE, 2012.
6. Saha, Sumit, et al. "Energy Efficient LIF Neuron Circuit Using Hybrid CMOS-NEMS in 65 Nm CMOS Technology." *2022 IEEE 35th International Conference on Micro Electro Mechanical Systems Conference (MEMS)*. IEEE, 2022.
7. Calimera A, Macii E, Poncino M. *The Human Brain Project and neuromorphic computing*. *Funct Neurol*. 2013 Jul-Sep;28(3):191-6. PMID: 24139655; PMCID: PMC3812737.
8. Indiveri, Giacomo, and Timothy K. Horiuchi. "Frontiers in neuromorphic engineering." *Frontiers in neuroscience* 5 (2011): 118.
9. Zenke, Friedemann, et al. "Visualizing a joint future of neuroscience and neuromorphic engineering." *Neuron* 109.4 (2021): 571-575.
10. Pham, Martin Do, et al. "From Brain Models to Robotic Embodied Cognition: How Does Biological Plausibility Inform Neuromorphic Systems?." *Brain Sciences* 13.9 (2023): 1316.
11. Lepri, N., et al. "In-memory computing for machine learning and deep learning." *IEEE Journal of the Electron Devices Society* (2023).

5. Can author be more specific about what part is done by experiment and what part is done by simulation in all their application demonstrations (Figs. 4-5)? More details on how the measurements in the application demonstrations are performed should be provided.

We thank the Reviewer for bringing to our attention that this aspect was unclear, and we apologize for not adequately addressing the demarcation between experimental and simulation study in the original manuscript.

Frequency tonotopic sensing on a logarithmic scale is experimentally demonstrated in Fig. 3. To further exploit the potential of tonotopic sensing, the systems of Fig. 4 and 5 are simulated employing the probabilistic model reported in Supplementary Note 10. The parameters used in these probabilistic models are carefully derived from the experimental characterization of the devices and validated as reported in Supplementary Note 11, following the characterization procedure for switching probability reported in the "Stochastic switching of volatile RRAM devices" section. These simulations of application demonstrations serve as compelling proof of concept, illustrating the remarkable capabilities achievable through the utilization of tonotopic sensing, experimentally demonstrated in Fig.3, in novel and groundbreaking ways.

In fact, it should be noticed that the applications of cochlea-inspired tonotopic sensing and speech recognition inevitably result in higher complexity in terms of an increased number of devices. Nonetheless, the fundamental behavior of these systems is rooted in the tonotopic mapping

introduced and experimentally validated in Fig. 3. This constitutes the heart of the system, where the addition of XOR gates simply acts as a means of transforming data according to one-hot encoding as reported in Supplementary Note 12. The increased number of devices is employed to efficiently tackle advanced tasks such as speech recognition.

To better highlight the separation between the experimental and simulation studies, we modified the main text in the sections “RRAM circuit for frequency sensing”, “Cochlea-inspired tonotopic sensing of audio frequency” and “Interpretability of tonotopic map: speech recognition” (pp. 5-8).

REVIEWERS' COMMENTS

Reviewer #1 (Remarks to the Author):

The authors had addressed all the concern.

Reviewer #2 (Remarks to the Author):

The authors have meticulously addressed all of my concerns in their comprehensive responses to the questions. Their 32-page response letter, totaling 10,000 words, reflects a thorough and detailed examination of the issues. I find their response to be both detailed and elaborate. After reviewing the revised manuscript, I have no further comments to add and am pleased to recommend it for publication.

Reviewer #3 (Remarks to the Author):

The author has made significant efforts to include additional information addressing the reviewer's queries. The quality of the manuscript has improved substantially. Therefore, I recommend that this manuscript be accepted for publication.

REVIEWERS' COMMENTS

Reviewer #1 (Remarks to the Author):

The authors had addressed all the concern.

Reviewer #2 (Remarks to the Author):

The authors have meticulously addressed all of my concerns in their comprehensive responses to the questions. Their 32-page response letter, totaling 10,000 words, reflects a thorough and detailed examination of the issues. I find their response to be both detailed and elaborate. After reviewing the revised manuscript, I have no further comments to add and am pleased to recommend it for publication.

Reviewer #3 (Remarks to the Author):

The author has made significant efforts to include additional information addressing the reviewer's queries. The quality of the manuscript has improved substantially. Therefore, I recommend that this manuscript be accepted for publication.

RESPONSE TO REVIEWERS

We thank the reviewers for the revision work done on our manuscript, which has allowed us to improve its quality.